# Few-Shot Detection of Machine-Generated Text using Style Representations

**Rafael Rivera Soto**[1,3,†], **Kailin Koch**[1], **Aleem Khan**[3],
**Barry Chen**[1], **Marcus Bishop**[2], **Nicholas Andrews**[3,†]
[1]Lawrence Livermore National Laboratory
[2]U.S. Department of Defense    [3]Johns Hopkins University

## Abstract

The advent of instruction-tuned language models that convincingly mimic human writing poses a significant risk of abuse. However, such abuse may be counteracted with the ability to detect whether a piece of text was composed by a language model rather than a human author. Some previous approaches to this problem have relied on supervised methods by training on corpora of confirmed human- and machine-written documents. Unfortunately, model under-specification poses an unavoidable challenge for neural network-based detectors, making them brittle in the face of data shifts, such as the release of newer language models producing still more fluent text than the models used to train the detectors. Other approaches require access to the models that may have generated a document in question, which is often impractical. In light of these challenges, we pursue a fundamentally different approach not relying on samples from language models of concern at training time. Instead, we propose to leverage representations of writing style estimated from human-authored text. Indeed, we find that features effective at distinguishing among human authors are also effective at distinguishing human from machine authors, including state-of-the-art large language models like `Llama-2`, `ChatGPT`, and `GPT-4`. Furthermore, given a handful of examples composed by each of several specific language models of interest, our approach affords the ability to predict which model generated a given document. The code and data to reproduce our experiments are available at https://github.com/LLNL/LUAR/tree/main/fewshot_iclr2024.

## 1 Introduction

Recent interest in large language models (LLM) has resulted in an explosion of LLM usage by a wide variety of users. Although much of this usage may be well-intentioned and benign, a growing concern is the usage of LLM for deception, such as for phishing attacks, disinformation, spam, and plagiarism (Hazell, 2023; Weidinger et al., 2022).

To minimize the risk of abuse of *commercial* systems, one recently proposed recourse is for those systems to employ statistical *watermarking* techniques (Kirchenbauer et al., 2023). Although watermarking may help mitigate some unintended consequences of LLM adoption, the advent of open-source LLM with performance approaching that of commercial LLM and achievable on commodity hardware raises the possibility of circumventing watermarking mechanisms to generate harmful content, potentially at a large scale. Thus, watermarking fails to completely address the issue of malicious content.

One reasonable recourse is to develop automatic detection approaches that attempt to predict whether a given document was composed by an LLM rather than by a human author. To this end, a number of methods have been proposed, such as OpenAI's AI Detector (Solaiman et al., 2019). An obvious drawback of classification methods, particularly those based on deep learning, is that the resulting models are susceptible to shifts in data distribution between training and deployment time for a variety of reasons, such as limited training data or the non-stationary of text (D'Amour et al., 2022). Furthermore, prior work in machine-text detection has found that in order to achieve good

---

†Corresponding authors: rafaelriverasoto@jhu.edu, noa@jhu.edu

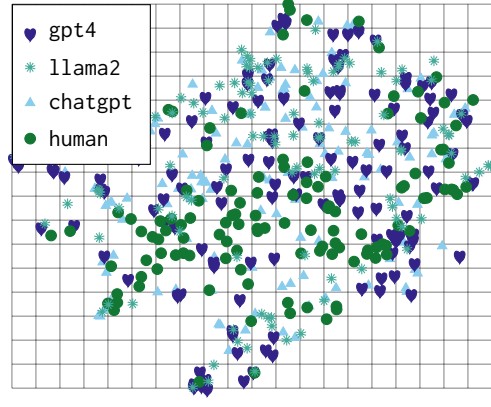 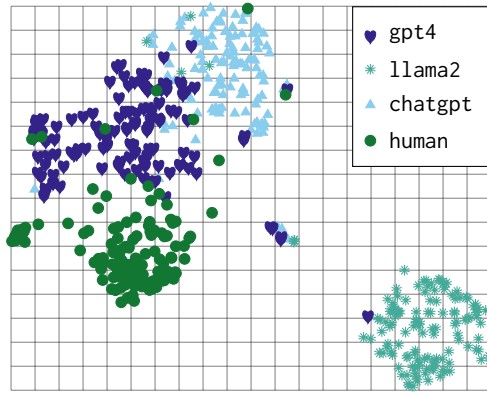

(a) Semantic document embeddings  (b) Stylistic document embeddings

Figure 1: UMAP projections (McInnes et al., 2018) of semantic or stylistic representations of writing samples in the Reddit domain composed by human or machine authors. We use SBERT as a representative dense semantic embedding (Reimers and Gurevych, 2019) and UAR as a representative stylistic representation (Rivera-Soto et al., 2021). Each point shown is the result of embedding a document containing at most 128 subword tokens for a standard vocabulary of size 50K. Despite using prompts designed to elicit a variety of writing styles from the LLM, the stylistic representation separates human from machine authors and machine authors from one another significantly better than the semantic representation.

performance from such a classifier, it is necessary to train the model with documents generated by the specific LLM one wishes to detect (Zellers et al., 2019). As a result, such models need to be updated whenever new LLM are introduced, something which could be both impractical and expensive in light of the frequent release of ever-improving models.

Human writing is characterized by a wide variety of stylometric features. In this paper, we explore the question of whether LLM exhibit consistent writing style across a range of prompts, particularly when explicitly prompted to generate text mimicking specific styles. In order to make the notion of writing style more concrete, we propose to employ representations of style learned from large corpora of writing samples by human authors that aim to capture invariant features of authorship (Wegmann et al., 2022; Rivera-Soto et al., 2021). Figure 1 illustrates that such representations do indeed separate documents composed by human authors from those composed by LLM. Moreover, we also observe that text generated by any particular LLM follows a style distinct from the styles of human authors and also distinct from those of other LLM. Thus, writing style affords the ability to detect not only that a given document was composed by an LLM, but also to predict which LLM generated it, given only a few example documents composed by each of several LLM of concern. The proposed approach may therefore reveal that specific LLM are being abused, something which increases transparency and accountability for companies disseminating LLM without appropriate controls or safeguards.

In light of the unavoidable distribution shifts stemming from the introduction of new LLMs, topics, and domains, this work focuses on the few-shot setting. Specifically, our evaluations assess the ability to detect writing samples produced by LLM *unseen* at training time, and in some cases, drawn from new domains and dealing with new topics. Our approach outperforms prominent few-shot learning methods as well as standard zero-shot baselines and differs significantly from prior work in that we do not require access to the predictive distribution of the unseen LLM, like Mitchell et al. (2023), or a large number of samples from it, like Zellers et al. (2019), to effectively detect text generated by these models.

We also explore factors leading to effective style representations for this task, finding that contrastive training on large amounts of human-authored text is sufficient to obtain useful representations, but that in certain few-shot settings, training on additional LLM-generated documents significantly improves performance. Finally, we release the datasets we generated as part of this work, which include documents generated by a variety of language models.

## 2 RELATED WORK

Perhaps the most widely-applied machine-text detector is OpenAI's AI Detector, which is intended to predict whether a given document was human- or machine-generated (Solaiman et al., 2019). The model was trained using documents generated by GPT-2 comprising one class and documents drawn from the corpus used to train GPT-2 comprising the other class. OpenAI released a similar classifier for detecting ChatGPT in January 2023. As of this writing, OpenAI has withdrawn the AI Detector, citing its low rate of accuracy. Indeed, it is well-known that supervised detectors may overfit various properties of their training data, such as specific decoding strategies (Ippolito et al., 2020).

By perturbing the output logits of an LLM and thereby its decoded text, a recent proposal known as *watermarking*, one may effectively detect text generated by LLM in some settings. For example, Kirchenbauer et al. (2023) encode a watermark in generated text by splitting a model's vocabulary into so-called *red* and *green* lists, encouraging tokens from the green list to be sampled during decoding more frequently than tokens from the red list. Because this line of work requires direct access to a model and its vocabulary, the approach is most relevant for models deployed by an organization through an API (He et al., 2022). However, adversaries may decline to generate watermarked text, for example, by using their own LLM, or remove the watermark from API-generated text, for example, by paraphrasing, which has emerged as an effective mechanism to circumvent the detection of watermarked text (Krishna et al., 2023). We conduct a study comparing the proposed approach to watermarking in Appendix E.

Much of the recent work on detecting machine-generated text has focused on training a classifier using datasets of human- and machine-generated text (Jawahar et al., 2020), requiring access to a model's predictive distribution for comparison with other distributions (Mitchell et al., 2023), or directly using a model to detect its own outputs (Zellers et al., 2019). Other work has looked at patterns of repetition typical of LLM to rank documents according to their likelihoods of being machine-generated (Gallé et al., 2021). In other recent work, the reliability of machine-text detectors in general has been questioned on theoretical grounds (Sadasivan et al., 2023). Zero-shot detection of machine-generated text is also a difficult task for human discriminators. Indeed, Dugan et al. (2020) demonstrated that human annotators had difficulty pinpointing changepoints between passages of documents composed by humans and those composed by machines, while Maronikolakis et al. (2021) find that automatic detectors can outperform humans in certain settings.

Given the limitations of discriminative classifiers at detecting machine-written documents, one might wonder whether generative approaches would be more robust. Unfortunately, deep generative models are also vulnerable to distribution shifts in general (Nalisnick et al., 2019), whereas effective machine-generated text detectors must be robust to shifts in topic, domain, and sampling techniques relative to those employed at training time. Our proposed approach addresses this concern by using learned stylistic representations, which are trained to ignore features that evolve over time, such as topic and domain, and focus on stylistic features that authors use more consistently (Rivera-Soto et al., 2021; Wegmann et al., 2022).

## 3 METHODS

**Detection regimes** The experiments in this paper deal with both the supervised and few-shot learning regimes. In the supervised setting, we avail of a training corpus consisting of human- and machine-written documents. The goal is to estimate a score proportional to the likelihood that a held-out document was generated by an LLM. In contrast with the few-shot setting described below, here we assume that we have at least hundreds but preferably thousands or more examples composed by a variety of language models. In practice the likelihood estimate is implemented by fine-tuning an underlying, pre-trained feature representation. We consider two such representations in this paper, namely *semantic representations*, where features are obtained by fine-tuning pre-trained representations based on masked language models, and *style representations*, where features are obtained by fine-tuning pre-trained style representations, which we discuss in more detail below. All being well, the resulting classifier would generalize to various test conditions, including novel language models, topics, and domains, a question we examine in Appendix A.

In the few-shot setting we assume that for each LLM of concern we have a *support sample*, which consists of a small number of *in-distribution* documents composed by that model. For example,

upon recognizing by manual inspection that some of their students' assignments exhibit telltale signs of machine-generation, such as hallucination, an instructor could use those essays in question as the support sample. Alternatively, the instructor might proactively prepare the support sample by prompting various LLM themself. The goal in this setting is to estimate the likelihood that one of the LLM in question generated a given document, and if so, a secondary objective is to predict which model generated the document.

**Style representations**   A primary component of our proposed approach to detecting machine-generated text is the notion of a *style representation*, something we employ to help our detectors overcome the challenges of generalization to new LLM, topics, and domains. We denote a style representation generically by $f$, which is taken to be some auxiliary model mapping a handful of documents $x_1, x_2, \ldots, x_K$ to a fixed-dimensional vector $f(x_1, x_2, \ldots, x_K)$, the *representation* of that document collection. The mapping $f$ is fit such that $f(x_1, x_2, \ldots, x_K)$ and $f(x'_1, x'_2, \ldots, x'_L)$ have large cosine similarity if and only if the writing style of $x_1, x_2, \ldots, x_K$ is similar to that of $x'_1, x'_2, \ldots, x'_L$.

We observe that writing style often comes into focus only after observing a sufficiently-large writing sample. For example, the repeated usage of a rare word may be discriminative of a particular author, but observing repeated word usage typically requires observing more than a few sentences. Indeed, our best results are obtained by passing longer spans of text to the model by combining style representations of multiple short documents using a learned aggregation mechanism.

In light of the highly nuanced nature of writing style, learning generalizable style representations requires large amounts of data. To this end, prior work has leveraged the availability of proxy author labels available in the form of account names on various social media platforms, such as blogs, microblogs, technical fora, and product reviews. In this paper, we focus on Reddit posts, which provide writing samples from authors discussing a wide variety of topics in various communities, which are known as *subreddits*. Furthermore, these topics cover diverse author interests and backgrounds, resulting in a corpus representing diverse styles. Our datasets are described in more detail in §4.1.

**Training style representations**   One of the primary challenges in learning style representations is to disentangle invariant features of writing, such as style, from features that vary over time, such as topic. To achieve this, we employ the following contrastive training strategy. Assuming our training dataset includes histories of each contributing author's writing spanning several months or years, we pair writing samples composed at different points in time by the same author to yield *positive examples*, and pair writing samples by different authors to yield *negative examples* (Andrews and Bishop, 2019). As a point of comparison, we additionally include results using style representations estimated using the approach of Wegmann et al. (2022), which differs in two significant ways. First, *non-episodic* training is used, meaning that features are computed of individual documents rather than *episodes* of multiple documents. Second, the approach relies on topic annotations, namely the *subreddit* feature, to construct *hard positives* by pairing writing samples composed by a single author discussing different topics, and *hard negatives* by pairing samples composed by different authors discussing similar topics.

**Proposed few-shot method**   An author-specific style representation $f$ admits a straightforward mechanism for few-shot detection. Specifically, suppose $x_1, x_2, \ldots, x_K$ is a handful of documents known to have been generated by a particular language model of interest, where we regard a *document* to be short span of text of around the length of a social media post. Specifically, unless otherwise specified, all documents considered in this work have length around 128 tokens according to a fixed tokenizer. Given a handful of new of documents $x'_1, x'_2, \ldots, x'_L$ at inference time, we compute the cosine similarity between $f(x_1, x_2, \ldots, x_K)$ and $f(x'_1, x'_2, \ldots, x'_L)$ to obtain a score monotonically related to the estimated likelihood that $x'_1, x'_2, \ldots, x'_L$ were composed by the same source as $x_1, x_2, \ldots, x_K$. The score may be further *calibrated* to yield a meaningful confidence estimate, say, by using Platt Scaling (Platt et al., 1999). Otherwise these scores must be *thresholded* to yield same-author predictions.

In our experiments we focus on UAR, a RoBERTa-based architecture trained with a supervised contrastive objective according to the recipe described in Rivera-Soto et al. (2021). We use the reference implementation accompanying that work. We observed in preliminary experiments that

increasing the number of human authors contributing to the dataset used to train UAR improves performance on the authorship attribution task and improves generalization. Therefore, we train UAR using a larger dataset than in prior work, namely a corpus of Reddit comments composed by five million authors that we describe in §5. In addition to this instance of UAR, we also prepare the following variations.

**Multi-LLM variation** In our first variation, we initialize the model weights of UAR to those of the instance above trained on the comments of five million Reddit users. We continue training the model using posts by human authors drawn from the `r/politics` or `r/PoliticalDiscussion` subreddits, as well as posts to these subreddits generated by LLM. In §4.1 we discuss the generation procedure and the motivation for controlling the topic by restricting to these two subreddits. Because training UAR involves optimizing a contrastive objective on each pair of episodes drawn from the same training batch, we ensure that half of the episodes in each batch originate from human authors and half from LLM. In this setting we regard a language model as an *author* distinct from other members of its LLM family. For example, `GPT2-large` is taken to be distinct from `GPT2-xl`.

**Multi-domain variation** Previous work has shown that including multiple domains during training can improve the quality of style representations (Rivera-Soto et al., 2021). To this end, we prepare a variation of UAR by augmenting the comments of five million Reddit users with data drawn from Twitter and StackExchange, ensuring that each training batch contains contributions from all three domains. See Table 3 of Appendix C for further statistics of this augmented dataset.

## 4 EXPERIMENTS

### 4.1 DATASETS

We distinguish between two kinds of language models, namely *amply available and cheap* (AAC) models, and models that users are *likely to want to detect* (LWD). We use documents generated by AAC models to estimate or fine-tune the feature representations described in §3 and hold out documents generated by LWD models for evaluation. This framework is intended to simulate the emergence of new LLM with more powerful capabilities, including the ability to better mimic human authorship.

To create the AAC dataset, we generate documents using readily available and computationally inexpensive models, namely `GPT-2` (Radford et al., 2019) and `OPT` (Zhang et al., 2022). For `GPT-2` we use the `large` and `xl` variants, which have 774M and 1.5B parameters respectively. For `OPT` we use two variations that have 6.7B and 13B parameters respectively. We generate text using human-generated posts to `r/politics` and `r/PoliticalDiscussion` as input prompts. By restricting to subreddits dealing with politics, a topic that was chosen for the substantial number of documents dealing with it, we aim to control for topic, thereby introducing an inductive bias encouraging representations learned from this dataset to separate machine and human authors on the basis of features unrelated to topic. All prompts contain at least 64 tokens according to the `GPT-2` tokenizer. We generate completions using a variety of parameters described in Table 4a of Appendix C. In total, we consider 64 combinations of decoding strategy, decoding value, temperature parameter, subreddit prompt source, and model size. After generating the completions, we truncate each document to the last sentence boundary before the 128th token, where we identify sentence boundaries using `spaCy` (Honnibal and Montani, 2017). Controlling for topic and length is intended to ensure that models trained with this corpus cannot easily distinguish between documents generated by human or machine authors by learning extraneous features, namely topic and length.

We evaluate our proposed detection approaches using the LWD dataset that we now describe, as well as M4 (Wang et al., 2023), a recently-released dataset containing documents generated by multiple LLM in five domains. We construct the LWD dataset by generating text with `Llama-2`, `GPT-4`, and `ChatGPT` using prompts of the form `write an Amazon review in the style of the author of the following review: ⟨human review⟩`, where ⟨human review⟩ is a real Amazon review. Prior work has shown that LLM have some ability to reproduce styles provided through in-context examples (Reif et al., 2022; Patel et al., 2023). Thus, our data generation procedure aims to induce stylistic variety to the extent currently possibly by state-of-the-art LLM, with the goal of increasing the difficulty of our benchmark. More details on the LWD and M4 datasets may be found in Appendix C.

## 4.2 Few-shot and zero-shot detection baselines

We compare our proposed few-shot approach to Prototypical Networks (Snell et al., 2017) and MAML (Finn et al., 2017), both prepared using an underlying RoBERTa model and trained with the AAC dataset described in §4.1 together with human-generated posts to `r/politics` and `r/PoliticalDiscussion`, where each distinct LLM contributing to the AAC corpus is regarded as a single author. To confirm that the controls for topic and length discussed in §4.1 are also helpful for our baseline approaches, we report additional ablation experiments on these choices in Appendix B. Next, we introduce variations of our proposed approach in which we replace the UAR representation with alternative embeddings, namely SBERT (Reimers and Gurevych, 2019)[1] and CISR (Wegmann et al., 2022)[2], the latter being a further authorship representation that leverages hard negatives and hard positives at training time. Additionally, we introduce several zero-shot baseline models, including two versions of the OpenAI Detector (Solaiman et al., 2019), one off-the-shelf and one that we train ourselves using the AAC corpus and Reddit politics posts. Finally, we include detectors based on metrics derived from `GPT2-xl` likelihood predictions, including `Rank` (Gehrmann et al., 2019), `LogRank` (Solaiman et al., 2019), and `Entropy` (Ippolito et al., 2020).

## 4.3 Metrics

We use the Receiver Operating Characteristic (ROC) curve to assess detection performance as the corresponding detection threshold varies. To summarize the ROC curve and compare different methods across operating points, we report the standardized partial area under the ROC curve restricted to the range of operating points corresponding with false alarm rates not exceeding 1%, which we denote by pAUC in this work. This allows us to better compare high-performing systems, noting that the proposed few-shot learning methods achieve nearly-perfect scores when calculating the area under the entire ROC curve, even when supplied with very few examples generated by LLM of concern. Another rationale for using pAUC is that readers worried that users may mistrust systems that raise too many false alarms will be most interested in the range of operating points corresponding with low false-alarm rates. This is intended to help readers arrive at operating points that minimize time-consuming followup inspection. However, we conduct similar experiments in Appendix F using smaller writing samples and for which we report additional metrics, including the usual AUC and FPR@95. We calculate pAUC using the `roc_auc_score` function from `scikit-learn` (Pedregosa et al., 2011) by specifying the parameter `max_fpr=0.01`.

## 4.4 Single-target machine text detection

In the experiment reported in this section, we assume access to a small *in-distribution* writing sample generated by a *specific* LLM of concern, such as `ChatGPT`. The objective is to identify further writing samples generated by this same model from among the documents in a large collection. This evaluation reflects the setting where one would like to perform targeted detection of a particular LLM. For example, an instructor might like to be alerted to cases where their students may have used a particular LLM as a writing assistant. In §4.5 we evaluate the same detection approach in the setting where the task is to identify documents generated by *any* of *multiple* LLM.

We evaluate all detection approaches considered using the evaluation corpus described in §4.1, which contains documents composed by human authors or LWD models in five domains, namely the Amazon documents we prepared and the documents from M4 drawn from domains other than Reddit, which serves as the primary training domain. All documents are grouped into *episodes* of $N$ documents by the same human or machine author for the desired value of $N$, each document containing a maximum of 128 tokens and ending at a sentence bounadary.

We apply the following procedure for each detection approach considered and each desired value of $N$. For each evaluation domain and each LLM contributing episodes in that domain, we take each episode generated by that LLM in turn to serve as the *support sample*, with all remaining episodes generated by that LLM and all human-generated evaluation episodes serving as *queries*. However, in the case of the MAML approach we take only the first 20 episodes by the LLM to serve as support samples due to computational burden. We calculate a score for every query indicating how similar it

---

[1] `https://huggingface.co/sentence-transformers/paraphrase-distilroberta-base-v1`
[2] `https://huggingface.co/AnnaWegmann/Style-Embedding`

is to the support sample, noting that the way this score is calculated depends on the detection method being evaluated. Each score is paired with a label indicating whether the query was composed by the same LLM as the support sample or was composed by a human author. Finally, we calculate a ROC curve based on these scores and labels and its corresponding pAUC value.

The results of this calculation are shown in Table 1, which reports the mean and standard error over all support samples, all evaluation domains, and all LLM for each detection approach considered and each value of $N$. The proposed method based on representations of writing style outperforms all other approaches. We note that the ProtoNet detector is trained on the AAC corpus and has 40M more parameters than UAR, so the superior performance of the style representation methods is not a consequence of larger model capacity. In Appendix D we break down these results according to evaluation domain.

| Method | Training Dataset | pAUC | |
| --- | --- | --- | --- |
| | | $N = 5$ | $N = 10$ |
| **Few-Shot Methods** | | | |
| UAR | Reddit (5M) | **0.905 (0.001)** | **0.9806 (0.0006)** |
| UAR | Reddit (5M), Twitter, StackExchange | 0.886 (0.001) | 0.9676 (0.0008) |
| UAR | AAC, Reddit (politics) | 0.877 (0.001) | 0.9400 (0.0013) |
| CISR | Reddit (hard negatives, positives) | 0.839 (0.001) | 0.9331 (0.0013) |
| ProtoNet | AAC, Reddit (politics) | 0.871 (0.001) | 0.9475 (0.0014) |
| MAML | AAC, Reddit (politics) | 0.662 (0.006) | 0.6854 (0.0068) |
| SBERT | Multiple | 0.621 (0.002) | 0.7157 (0.0022) |
| **Zero-Shot Methods** | | | |
| AI Detector (custom made) | AAC, Reddit (politics) | 0.6510 (0.031) | 0.6585 (0.0320) |
| AI Detector (off-the-shelf) | WebText, GPT-xl | 0.6028 (0.0250) | 0.6011 (0.0249) |
| Rank | BookCorpus, WebText | 0.5693 (0.0152) | 0.5581 (0.0172) |
| LogRank | BookCorups, WebText | 0.7640 (0.0360) | 0.7749 (0.0378) |
| Entropy | BookCorpus, WebText | 0.4984 (0.0005) | 0.4977 (0.0002) |
| Random | | 0.5 | 0.5 |

Table 1: Single-target detection results. Each model was evaluated on a common corpus of documents involving unseen domains, topics, and LLM, organized into episodes of $N$ documents. The standard errors shown in parenthesis were estimated using bootstrapping.

To study the effect of the number $N$ of documents comprising each episode, we vary $N$ between 1 and 10, still truncating each document to the nearest sentence boundary before the $128^{th}$ token. The results are shown in Figure 2a. We observe that the version of UAR trained only with Reddit performs best across the range, although other UAR variants perform well also. Note that stylistic representations do not require AAC data during training to capture the style of machine generators. Indeed, both UAR and CISR outperform methods that require AAC training data.

We also observe that metric-based approaches like ProtoNet outperform fast-adaptation approaches like MAML, in which a new model is fitted to each support sample in turn. One reason for this may be that MAML is limited to inputs of 512 tokens at inference time due to its underlying RoBERTa model, which was chosen to match the underlying models used by other baseline approaches. In contrast, metric-based approaches may combine representations of various spans of text together, thus effectively increasing the context from which they make predictions. Another reason for this discrepancy may be that fast-adaptation approaches are more prone to over-fitting the support sample.

## 4.5 MULTIPLE-TARGET MACHINE TEXT DETECTION

In §4.4 we handled the case of detecting a *single* target language model. We now extend our formulation to include *multiple* target language models, given support samples from each model. Namely, for each detection approach and each $N$ we perform the following calculation. For each domain we repeat the following trial 1000 times. For each LLM contributing to the evaluation corpus we randomly select a support sample composed by that LLM and take all remaining evaluation episodes to serve as queries. For each query, we calculate a score for each support sample reflecting its similarity to the query, and pair the *minimum* score with a label indicating whether the query was composed by a human or an LLM. Finally, we calculate the pAUC based on these minimum scores

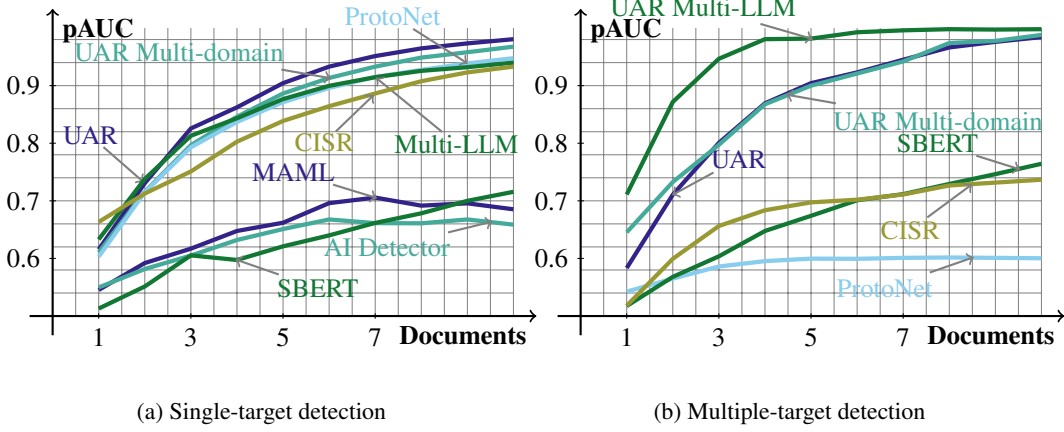

(a) Single-target detection        (b) Multiple-target detection

Figure 2: Detection performance as the number of documents $N$ comprising episodes varies.

and labels. The results in Figure 2b show the mean pAUC over all domains, all LLM, and all trials against $N$ for each detection approach.

We find that in this setting, the proposed approach with the multi-LLM authorship model performs best, although the other UAR variants remain competitive, noting that this model is the only authorship model considered which is explicitly trained to distinguish among LLM, including different versions of the same LLM. We also note that the evaluation corpus contributing most to the difference is the PeerRead component of M4, as shown in Appendix D, where we break down the results of this experiment by evaluation dataset. Otherwise the results are similar for all evaluation domains.

Note that in this experiment, a key assumption is that each support sample is known to have originated *entirely* from one of several LLM of concern. However, it may be possible, say, by manual inspection, to ascertain that each of a handful of documents was generated by *some* LLM, but not necessarily all by the *same* LLM. This setting is addressed in Appendix G.

## 4.6 ROBUSTNESS AGAINST PARAPHRASING ATTACKS

We now test the effectiveness of our proposed approach against an adversary that applies automatic paraphrasing to LLM-generated text to evade detection. To simulate this scenario, we paraphrase various evaluation episodes using DIPPER (Krishna et al., 2023) with a lexical diversity parameter of 20% as described below. All episodes in this experiment consist of $N = 5$ documents. We first repeat the procedure described in §4.4 using the version of UAR trained on the comments of 5M Reddit users, except that we now paraphrase a varying proportion of the episodes generated by the target LLM other than the support sample. The results of this experiment are labeled by *UAR Single-target* in Figure 3, along with the results of the same experiment using the ProtoNet baseline. Unfortunately, both approaches suffer as the proportion of queries paraphrased increases, but we observe that like the adversary, the detector may also avail of paraphrasing! To this end, we repeat the procedure described in §4.5 with the following modification. For each LLM contributing to the evaluation corpus and each episode generated by that model serving as the support sample, we paraphrase a varying proportion of the remaining episodes generated by that model and report the minimum of *two* scores: one reflecting the likelihood that the support sample matches a randomly selected query, and another reflecting the likelihood that a *paraphrase* of the support sample matches a randomly selected query. The results of this experiment are labeled by *UAR Multi-target* in Figure 3. Indeed, including a paraphrase of the support sample mitigates the drop in performance.

## 5 CONCLUSION

**Summary of findings** We propose a few-shot strategy to detect machine-generated text using style representations estimated from content primarily composed by humans. Our main finding is that style representations afford a remarkable ability to identify instances of text composed by LLM

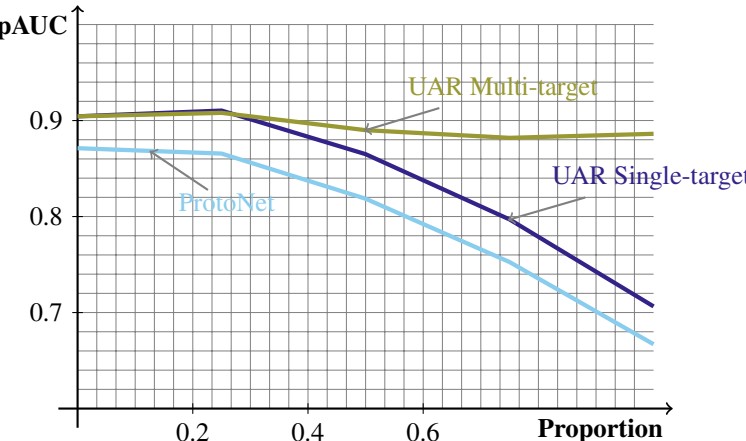

Figure 3: Mean of pAUC as the proportion of *queries* is paraphrased. Paraphrasing reduces the detection rate across the low FPR range, but including the paraphrased LLM as a support sample (UAR Multi-LLM) mitigates the drop in performance.

given only a handful of demonstration examples, even when those examples were generated with prompts engineered to elicit diverse writing styles. In addition, we illustrate in §4.5 that further improvements are possible for multiple-target LLM detection by fine-tuning style representations using documents generated by AAC models. Finally, in §4.6 we illustrate that the multiple-target variant of the proposed approach that incorporates paraphrasing is robust against paraphrasing attacks. By focusing our evaluation on the portion of the ROC curve corresponding with a false-alarm rate of less than 1%, we seek to emphasize that the proposed methodology represents a practically relevant approach to mitigating certain LLM abuses.

**Limitations**    The few-shot detection framework explored in this work assumes access to a handful of documents generated by LLM that users may wish to detect in a particular domain. As of this writing, there are only a small but growing number of such LLM, since only a handful of large companies have the resources to train such models. This makes it possible to anticipate which LLMs an adversary may abuse and proactively train detectors using *any* of the approaches considered in this work. However, in the future it seems plausible that a much larger number of LLM will be available, in which case a more reactive approach would be required, such as the approach proposed in this paper, where examples of abuse by unknown LLM are assembled to form the required support samples.

Our experiments focus on English since LLM of concern are primarily available in English. However, since the training procedure for the style representations used in our few-shot detection experiments relies only on the availability of author-labeled text, there are no barriers to developing such representations for arbitrary languages other than collecting sufficiently large corpora. We acknowledge that this is easier accomplished for high-resource languages, particularly those that are well-represented in online discourse. For these reasons, we believe that exploring style representations effective in low-resource languages is an interesting avenue for future work.

**Broader impact**    The rapid adoption and proliferation of LLM poses a risk of abuse unless methods are developed to detect deceitful writing. The proposed few-shot detection method represents a novel and practical approach to detecting machine-generated text in many settings, including plagiarism in classrooms, social media moderation, and email spam and phishing. The approach may be deployed immediately using readily available, pre-trained style representations and requires only a small number of examples generated by LLM of concern. We will release code and checkpoints of our best models to facilitate adoption of the proposed approach.

REPRODUCING OUR RESULTS

The proposed few-shot detectors were trained using a open-source reference implementation of UAR in PyTorch available at https://github.com/LLNL/LUAR using default hyperparameter choices. For the CISR baseline, we used the open-source PyTorch implementation available at https://github.com/nlpsoc/Style-Embeddings. For ProtoNet and MAML, we used the implementations provided at https://github.com/learnables/learn2learn. The data used to fine-tune the UAR style representations was sampled from a publicly available corpus of Reddit comments (Baumgartner et al., 2020). We subsampled this dataset for comments published between January 2015 and October 2019 by authors publishing at least 100 comments during that period. Additionally, we used Amazon reviews and StackExchange discussions in some model variations (Ni et al., 2019), both obtained from existing datasets. The Amazon dataset may be downloaded from https://nijianmo.github.io/amazon/index.html and the StackExchange dataset is available from https://pan.webis.de/clef21/pan21-web/style-change-detection.html. We also created two new corpora of machine-generated documents, referenced as AAC and LWD in the main text, which we used respectively for training and evaluation. In the case of AAC, we used publicly-released checkpoints for GPT-2 and OPT, available at the time of this writing from https://huggingface.co/models. In the case of LWD, we used the OpenAI API to generate documents with ChatGPT and GPT-4. We generated Llama2 examples using the llama2-7B chat model released July 2023, which can be found at https://github.com/facebookresearch/llama. We trained the style representations using one 8 x A100-80Gb GPU server, which took under 24 hours for each of the proposed model variations. Fop both UAR and CISR, the resulting style feature extractors have only 82M and 125M parameters respectively, and are therefore efficient to deploy on a single GPU.

ACKNOWLEDGMENTS

Part of this work was performed under the auspices of the U.S. Department of Energy by Lawrence Livermore National Laboratory under Contract DEAC52-07NA27344. This work was supported by the Office of the Director of National Intelligence (ODNI), Intelligence Advanced Research Projects Activity (IARPA), via the HIATUS Program under contract D2022-2205150003. The views and conclusions contained herein are those of the authors and should not be interpreted as necessarily representing the official policies, either expressed or implied, of ODNI, IARPA, or the U.S. Government. The U.S. Government is authorized to reproduce and distribute reprints for governmental purposes notwithstanding any copyright annotation therein.

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

## A GENERALIZATION OF SUPERVISED MACHINE-TEXT DETECTORS

The experiment reported in this section is intended to determine whether discriminative classifiers built on top of pre-trained style representations are more robust to future changes in topic, domain, and language model than those built on top of semantic representations. In order to simulate future scenarios, we train these models with text dealing with certain topics, drawn from certain domains, and with positive examples generated by older language models. Then we evaluate these models with text that may deal with new topics, may have been drawn from new domains, and with positive examples that may have been generated by newer language models. More information on these datasets, including some example documents we generated, may be found in Appendix C.

First we construct a supervised detector by simply composing an MLP with the UAR model pretrained on the Reddit comments of 5M users. We train this composition using equal amounts of human- and AAC-generated documents, keeping the parameters of UAR frozen. Next, mirroring the approach used by OpenAI's AI Detector, we compose an MLP with RoBERTa and train the composition using the same approach as the composition above.

The results are shown in Table 2a. Both approaches perform strongly when evaluated on data drawn from the same domain, dealing with the same topics, and generated by the same LLM as the training data. In fact, the RoBERTa baseline outperforms UAR, particularly at the lowest false positive rates, which is illustrated by the ROC curve shown in Figure 4a. However, when the same detectors are evaluated with data dealing with new topics or drawn from new domains, performance drops sharply, as shown in Figure 4b, Figure 4c, Figure 4d. In other words, the UAR+MLP combination is more robust to topic and domain shifts than the baseline. However, the performance of both detectors degrades to the point they would be unreliable as the FPR tends to zero. For example, at a FPR of $1\%$ both models reach a TPR of around $40\%$ on LWD datasets. This confirms our expectation about the limitations of using trained classifiers on unseen LWD data.

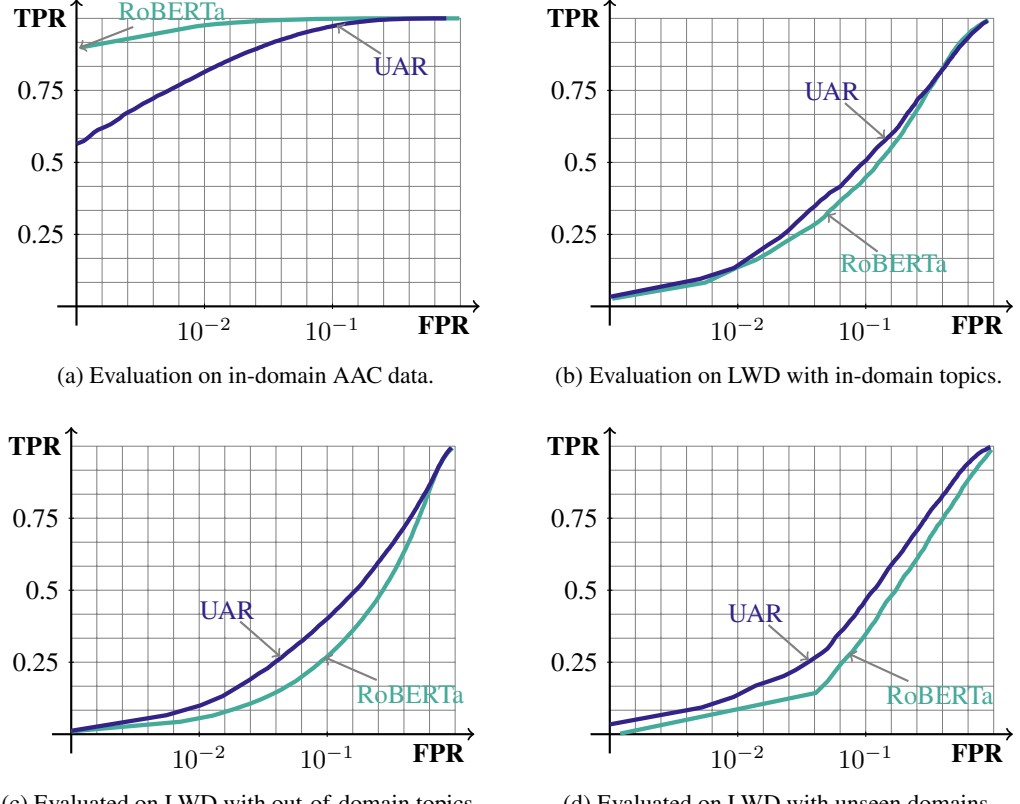

(a) Evaluation on in-domain AAC data.

(b) Evaluation on LWD with in-domain topics.

(c) Evaluated on LWD with out-of-domain topics.

(d) Evaluated on LWD with unseen domains.

Figure 4: ROC curves assessing supervised machine-text detection performance. Both the RoBERTa- and UAR-based detectors perform well in-distribution, but performance drops when evaluating on data generated by new LLM, new topics, or new domains. The UAR-based detector is more robust to changes in the testing distribution.

| Evaluation Set | UAR | RoBERTa |
|---|---|---|
| AAC | 0.8551 | 0.9671 |
| LWD | 0.5401 | 0.5363 |
| LWD, new topics | 0.5262 | 0.5136 |
| LWD, new domains | 0.5411 | 0.5007 |
| Random | 0.5 | 0.5 |

(a) pAUC scores for UAR and RoBERTa baseline on AAC and LWD.

| Ablation | pAUC |
|---|---|
| Control Topic and Length | 0.844 (0.009) |
| Control Topic Only | 0.769 (0.002) |
| No Control | 0.765 (0.002) |
| Random | 0.5 |

(b) Mean of pAUC across on Amazon. Each number is the mean of the distribution along with the standard error, estimated via bootstrapping.

Table 2: Further experimental results.

| Origin | Number of Authors | Number of Documents |
|---|---|---|
| StackExchange | 22,469 | 2,758,657 |
| Twitter | 1,905,705 | 370,277,856 |
| Reddit | 5,199,959 | 3,578,220,305 |

Table 3: Statistics of the human-generated datasets used to train the multi-domain authorship model.

## B PROTONET ABLATIONS

In order to understand the effect of controlling for topic and length in our AAC training data we perform the following ablation experiment. To control for length, we truncate each machine- and human-authored document to the last sentence boundary before the $64^{th}$ token. To control for topic, we ensure that all human-authored documents are drawn from only `r/politics` or `r/PoliticalDiscussion`. In the results shown in Table 2b we can see that when we control for topic only, the model is able to learn document length as a shortcut by which to distinguish human from machine authors, resulting in a relative decrease in pAUC of 8.8% at test time. Further ablating our control of topics has a less significant effect. The best results are obtained by controlling for both topic and length, which is the variation of ProtoNet we use in the main paper.

## C FURTHER DETAILS ABOUT DATASETS

Table 3 shows the numbers of authors and examples contributing to each of the human-generated training datasets we used to estimate the style representations used in our proposed few-shot detection method.

We now describe the process we used to create our AAC and LWD datasets. For both datasets, we used a single set of prompts to create a variety of documents. We balanced each dataset with an equal number of human-generated examples before splitting into training, validation and testing splits. We generated the AAC datasets using all possible combinations of the parameters specified in Table 4a. Some statistics of the resulting datasets are shown in Table 4b.

We used a similar approach to generate our LWD datasets. The models used were `GPT-4`, `ChatGPT` and `Llama-2`. We prompted each LWD model with a post sampled from either Reddit, Amazon, or

| Parameter | Values |
|---|---|
| Models | GPT2-large, GPT2-xl, OPT-6.7B, OPT-13B |
| Decoding Strategies | top-$p$, typical-$p$ |
| Decoding Values | 0.7, 0.95 |
| Temperature Values | 0.7, 0.9 |
| Generation Length | 512 tokens |

(a) Decoding parameters used to generate AAC datasets.

| Origin | Train | Valid | Test |
|---|---|---|---|
| Machine | 440,721 | 62,935 | 125,987 |
| Human | 440,721 | 62,935 | 125,987 |
| Total | 881,442 | 125,870 | 251,974 |

(b) Numbers of documents in datasets used to train and evaluate authorship models.

Table 4: Further details about AAC datasets.

| Dataset | Prompt Sources |
|---|---|
| AAC | `r/politics, r/PoliticalDiscussion` |
| LWD, same topic | `r/politics, r/PoliticalDiscussion` |
| LWD, different topic | `r/anime, r/MMA, r/movies, r/personalfinance, r/soccer` |
| LWD, different domains | Amazon product reviews, StackExchange posts |

Table 5: Prompt sources for LLM generation.

| LWD LMs | Same topic | New topic | New domain |
|---|---|---|---|
| Machine Generated Texts | 16,833 | 82,105 | 18,074 |
| Human Written Texts | 16,833 | 82,105 | 18,643 |
| Total Texts | 33,666 | 164,210 | 36,717 |

Table 6: Numbers of documents comprising various LWD splits.

StackExchange containing at least 64 tokens. For documents prompted with Reddit posts, we varied the prompts to elicit a diverse range of writing styles by specifying some personality traits of the supposed author. Some examples are shown in Table 7. For the documents prompted with Amazon and Stack Exchange posts, we prompted the language model to preserve the style of the post. Due to cost, we used GPT-4 to generate only Reddit content, but not Amazon or StackExchange.

Finally, in addition to our LWD dataset, we also used the recently-released M4 dataset (Wang et al., 2023) for evaluation. M4 consists of documents generated by multiple language models in multiple domains, including ArXiv, PeerRead, Reddit, WikiHow, and Wikipedia. Because only Reddit is used to prepare the style representations used in our experiments, we use a total of six unseen domains for evaluation. M4 includes documents generated by a variety of models, including ChatGPT (OpenAI, 2023), GPT-4 (OpenAI, 2023), Llama-2 (Touvron et al., 2023), Davinci(Brown et al., 2020), FlanT5 (Chung et al., 2022), Dolly (Conover et al., 2023), Dolly2 (Conover et al., 2023), BloomZ (Muennighoff et al., 2023), and Cohere. Table 9 shows the number of constituent documents from each domain.

## D  FURTHER VARIATIONS ON MAIN EXPERIMENTS

In Table 12 and Table 13 we break down the results of the experiments reported in §4.4 and §4.5 respectively by evaluation dataset. The relative performance of different approaches is largely consistent across evaluation datasets. We note that M4 PeerRead stands out as the most difficult corpora in both the single- and multiple-target scenarios. As noted by Wang et al. (2023), PeerRead exhibits fewer unique unigrams and bigrams, making it more difficult to separate human- from machine-authored texts on the basis of style alone.

## E  COMPARISON WITH WATERMARKING

Watermarking involves altering the token distribution of an LLM according to a fixed strategy (Kirchenbauer et al., 2023). Because this takes place when documents are generated, the approach assumes a benevolent adversary, a significant limitation relative to our proposed method. In this section we compare our approach to watermarking and also consider a simple mitigation that adversaries may deploy to circumvent watermarking.

We adapt the watermarking procedure outlined in the work cited above to apply watermarks to text generated by Llama-2 (Touvron et al., 2023). For this, we follow the same prompting procedure discussed in Appendix C to generate documents in the Amazon domain, which we aim to distinguish from real Amazon reviews using both our proposed approach and the statistical approach outlined in the reference above. The results in Figure 5 show that given a reasonable amount of text, our few-shot approach outperforms the statistical test.

However, an adversary may apply automatic paraphrasing to watermarked documents to make them less easy to detect. Recalling that we explored the robustness of the proposed few-shot approach to

| Persona | Example Document |
|---|---|
| Human Prompt | And the politicians that needed relief in Sandy were opposed to relief during Katrina.  And the politicians that needed relief during Katrina were opposed to relief for whatever disaster preceded it. Right now theres someone in, like, North Carolina reading about lawmakers who were opposed to Sandy relief but are now asking for relief in Louisiana, and theyre thinking Sure, this backfired on \*them\*, but I can safely vote against flood aid in Louisiana because theres no way this \*\*exact\*\* thing could happen to me! |
| Teenager | Ugh, it's just so frustrating how politicians only care about their own interests. It's like they forget that they're supposed to be serving the people and not just looking out for themselves. I wish they would stop playing politics with people's lives and actually do their jobs. |
| Concerned Citizen | As a concerned citizen, it's disheartening to see political leaders being hypocritical in times of need.  We need to prioritize humanity over political affiliations and offer aid to those who suffer from natural disasters regardless of their location or political beliefs. |
| Skeptical Optimist | I understand why you might feel skeptical about politicians who were once opposed to disaster relief suddenly asking for it when their own state is affected. However, as an optimist, I believe it's important to focus on the fact that these politicians are now recognizing the need for assistance during times of crisis. |
| Pragmatic Realist | As a pragmatic realist, I understand that politics often involves inconsistent and hypocritical behavior from lawmakers.  However, when it comes to disaster relief, it's important to separate politics from practicality. Regardless of a politician's stance on relief for previous disasters, the immediate needs of the current disaster should be addressed. |
| Passionate Activist | As an activist deeply committed to improving disaster relief policies, I find it appalling that politicians would maintain such hypocritical stances on disaster relief. We must recognize the need for and benefits of supporting our fellow Americans in times of crisis, regardless of their political affiliations or geographic location. |

Table 7: Example documents dealing with politics, generated by ChatGPT and prompted according to the desired personality of the author in order to elicit diverse writing styles.

| Dataset | Human Example | ChatGPT Example |
|---|---|---|
| Amazon | The kids immediately wanted to put on a puppet show with this (they received a box full of puppets separately), but no doubt this will also be used for a store and other things. | As an above-average fixed location-fixed view security camera, it provides SD quality images that can be conveniently viewed from the screen of a smart phone. |
| ArXiv Abstract | We introduce a density tensor hierarchy for open system dynamics, that recovers information about fluctuations lost in passing to the reduced density matrix. | The motivation for this research comes from the challenges encountered in simulating flows with strong nonequilibrium effects, such as flows with high-speed micro-jets, turbulent mixing, and multiphase flows. |
| Reddit ELI5 | For example, this is why there is a differentiation between being depressed (aka a depressive episode) and being diagnosed with major depressive disorder. | Now, as you may know, the Wright brothers - Orville and Wilbur - are widely credited with inventing the airplane. |
| Wikipedia | During the reign of the Shah kings, the Mulkajis (Chief Kajis) or Chautariyas served as prime ministers in a council of 4 Chautariyas, 4 Kajis, and sundry officers. | Born in Howard County, Maryland, on December 10, 1832, Carroll was the son of John Carroll, a prominent lawyer and politician from Maryland. |
| Wikihow | You will not always be the most intelligent person in the room, and the farther you get from school, the less book smarts will matter in your day-to-day life. | Whether you want to create a collage of memories for a special occasion or display your favorite photos in an artistic way, Inkscape can make it happen. |

Table 8: Examples of human text and ChatGPT generations from the Amazon and M4 datasets, truncated to a maximum of 32 tokens.

| M4 | Peerread | ArXiv Abstract | Reddit ELI5 | Wikihow | Wikipedia |
|---|---|---|---|---|---|
| Machine Generated Texts | 13,831 | 17,340 | 15,885 | 14,901 | 13,677 |
| Human Written Texts | 5,203 | 2,997 | 2,999 | 2,999 | 2,975 |
| Total Texts | 19,034 | 20,337 | 18,884 | 17,900 | 16,652 |

Table 9: M4 domains and statistics, noting that for evaluation we decline to use Reddit, which serves as our primary training domain.

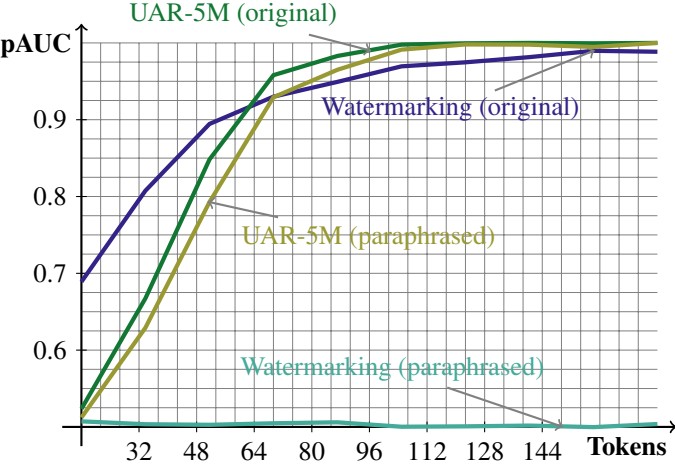

Figure 5: pAUC of the proposed approach and the watermark detector as the number of tokens is varied for the Amazon dataset. The proposed approach is more robust to paraphrase attacks, and achieves equal or better results to the watermark detector when the number of tokens is $\geq 48$.

paraphrasing in §4.6, we now compare the relative robustness of watermarking and our approach to the same paraphrasing attacks. We simulate this setting by using DIPPER to paraphrase each watermarked document, setting the lexical diversity parameter to 20%. We repeat the experiment above using the paraphrased documents in place of the original watermarked documents. The results shown in Figure 5 suggest that our approach is considerably more robust than the statistical watermarking test.

## F    EFFECT OF SHORTER TRUNCATION

In order to determine the effectiveness of the proposed approach when applied to smaller writing samples, we repeat the experiment described in §4.4, this time truncating all support samples and queries to the nearest sentence boundary before the $32^{\text{nd}}$ token. This is in contrast with the approach applied in the remainder of this work, where we truncated all samples and queries to the nearest sentence boundary before the $128^{\text{th}}$ token. We also report two additional metrics, namely the usual area under the ROC curve (AUC) and the false positive rate corresponding with a 95% true positive rate (FPR@95TPR). These results are reported in Table 10. Our proposed few-shot approaches continue to outperform baseline methods in this setting.

## G    DETECTION OF UNKNOWN LLM

Our final experiment follows the procedure described in §4.4 with the following modification. To create support samples, we randomly sample $N$ documents generated by LLM from the evaluation dataset. Thus, a support sample need not consist of documents by a single LLM, although the query episodes *do* consist of documents generated by a single author, either LLM or human. This experiment reflects the situation where a handful of documents are known to have originated from LLM, but the specific LLM generating each document cannot be attributed. As in §4.4, the detector score reflects the likelihood that a given query was generated by *any* LLM contributing to the evaluation corpus. The results are reported in Table 11.

We continue to see high detection accuracies in this detection setting, with the methods based on style representations outperforming other baselines. We now find that CISR performs better than other approaches, although the UAR variants remain competitive. Note that this experiment introduces a train-test mismatch for UAR, since each episode used to train UAR consists of documents by the same author, in contrast with the support samples used in the experiment, which typically contain documents by *multiple* LLMs.

| Method | Training Dataset | AUC | pAUC | FPR@95 |
|---|---|---|---|---|
| **Few-Shot Methods** | | | | |
| UAR | Reddit(5M) | **0.9884** | 0.9094 | **0.0533** |
| UAR | Reddit (5M), Twitter, StackExchange | **0.9884** | **0.9118** | 0.0543 |
| UAR | AAC, Reddit (politics) | 0.8913 | 0.7308 | 0.3394 |
| CISR | Reddit (hard negatives, positives) | 0.9608 | 0.7973 | 0.1440 |
| ProtoNet | AAC, Reddit (politics) | 0.9791 | 0.9062 | 0.0934 |
| MAML | AAC, Reddit (politics) | 0.6686 | 0.5141 | 0.6555 |
| SBERT | Multiple | 0.9673 | 0.8087 | 0.1448 |
| **Zero-Shot Methods** | | | | |
| AI Detector (custom) | AAC, Reddit (politics) | 0.7238 | 0.5295 | 0.5369 |
| AI Detector | WebText, GPT2-XL | 0.6933 | 0.5559 | 0.7698 |
| Rank | BookCorpus, WebText | 0.7301 | 0.5423 | 0.6341 |
| LogRank | BookCorpus, WebText | 0.9107 | 0.6395 | 0.2209 |
| Entropy | BookCorpus, WebText | 0.2083 | 0.4977 | 0.9667 |
| Random | | 0.5000 | 0.5000 | |

Table 10: Mean values of AUC, pAUC and FPR@95 for various detection approaches with episodes of $N = 10$ documents, each truncated to 32 tokens.

| Method | Training Dataset | pAUC | |
|---|---|---|---|
| | | $N = 1$ | $N = 2$ |
| UAR | Reddit (5M) | 0.682 (0.002) | 0.759 (0.002) |
| UAR | Reddit (5M), Twitter, StackExchange | 0.684 (0.002) | 0.706 (0.002) |
| UAR | AAC, Reddit (politics) | 0.640 (0.002) | 0.633 (0.002) |
| CISR | Reddit (hard negative, positives) | **0.707 (0.003)** | **0.779 (0.003)** |
| AI Detector (custom) | AAC, Reddit (politics) | 0.660 (0.029) | 0.668 (0.031) |
| ProtoNet | AAC, Reddit (politics) | 0.536 (0.001) | 0.524 (0.001) |
| MAML | AAC, Reddit (politics) | 0.672 (0.007) | 0.724 (0.010) |
| SBERT | Multiple | 0.552 (0.001) | 0.546 (0.001) |
| Random | | 0.5 | 0.5 |

Table 11: Results on detection of unknown LLM.

| Dataset | $N$ | UAR | UAR Multi-LLM | UAR Multi-domain | CISR | AI Detector (fine-tuned) | MAML | ProtoNet | SBERT | Random |
|---|---|---|---|---|---|---|---|---|---|---|
| LWD Amazon | 5 | 0.998 | 0.999 | 1.000 | 0.986 | 0.581 | 0.891 | 0.998 | 0.648 | 0.5 |
|  | 10 | 1.000 | 1.000 | 1.000 | 1.000 | 0.582 | 0.886 | 1.000 | 0.841 | 0.5 |
| M4 Arxiv | 5 | 0.989 | 0.951 | 0.890 | 0.996 | 0.659 | 0.702 | 0.951 | 0.653 | 0.5 |
|  | 10 | 1.000 | 0.990 | 0.975 | 1.000 | 0.641 | 0.731 | 0.995 | 0.713 | 0.5 |
| M4 PeerRead | 5 | 0.829 | 0.850 | 0.914 | 0.819 | 0.737 | 0.643 | 0.885 | 0.623 | 0.5 |
|  | 10 | 0.946 | 0.919 | 0.977 | 0.946 | 0.788 | 0.683 | 0.965 | 0.738 | 0.5 |
| M4 WikiHow | 5 | 0.871 | 0.828 | 0.862 | 0.659 | 0.499 | 0.538 | 0.799 | 0.659 | 0.5 |
|  | 10 | 0.980 | 0.920 | 0.964 | 0.814 | 0.499 | 0.539 | 0.918 | 0.757 | 0.5 |
| M4 Wikipedia | 5 | 0.845 | 0.777 | 0.801 | 0.753 | 0.718 | 0.670 | 0.747 | 0.519 | 0.5 |
|  | 10 | 0.979 | 0.878 | 0.930 | 0.919 | 0.715 | 0.700 | 0.866 | 0.566 | 0.5 |

Table 12: pAUC for single target detection experiment in §4.4, broken down by evaluation dataset.

| Dataset | $N$ | UAR | UAR Multi-LLM | UAR Multi-domain | CISR | ProtoNet | SBERT | Random |
|---|---|---|---|---|---|---|---|---|
| LWD Amazon | 5 | 0.999 | 0.999 | 1.000 | 0.999 | 0.995 | 0.597 | 0.5 |
|  | 10 | 1.000 | 1.000 | 1.000 | 1.000 | 1.000 | 0.699 | 0.5 |
| M4 Arxiv | 5 | 0.981 | 0.992 | 0.982 | 0.961 | 0.503 | 0.774 | 0.5 |
|  | 10 | 1.000 | 1.000 | 1.000 | 0.999 | 0.502 | 0.928 | 0.5 |
| M4 PeerRead | 5 | 0.665 | 0.994 | 0.733 | 0.500 | 0.500 | 0.559 | 0.5 |
|  | 10 | 0.935 | 0.998 | 0.974 | 0.517 | 0.499 | 0.705 | 0.5 |
| M4 WikiHow | 5 | 0.969 | 0.968 | 0.991 | 0.505 | 0.498 | 0.942 | 0.5 |
|  | 10 | 0.999 | 0.995 | 0.998 | 0.522 | 0.497 | 0.990 | 0.5 |
| M4 Wikipedia | 5 | 0.907 | 0.956 | 0.794 | 0.521 | 0.502 | 0.498 | 0.5 |
|  | 10 | 0.988 | 0.999 | 0.969 | 0.644 | 0.503 | 0.500 | 0.5 |

Table 13: pAUC for multi-target detection experiment in §4.5, broken down by evaluation dataset.

