# OpenReview forum: "Few-Shot Detection of Machine-Generated Text using Style Representations"
_ICLR.cc/2024/Conference — ICLR 2024 poster_

### Official Review · Reviewer_pncH · 2023-10-16

**Soundness:** 3 good
**Presentation:** 2 fair
**Contribution:** 2 fair
**Rating:** 6
**Confidence:** 4

**Summary:**

This paper presents a new algorithm for the task of AI-generated text detection. AI-generated text detection is a binary classification task of determining whether a given text was written by a human or a language model (like ChatGPT).

The key idea in this work is that the style of AI-generated text is different from human-written text, and this difference can be captured by a style representation model. First, the authors use a contrastive learning objective to learn style-dependent, but topic-independent representations. The authors leverage outputs from several cheap and readily available LMs (like GPT-2) to train the style representation model.

During inference, the authors assume access to a few exemplar samples of AI-generated text from each held-out language model that they consider. Given a candidate query, the authors measure the cosine similarity between the style representation of the query and the style representation of the exemplars. The authors then use a threshold on the similarity score to classify the query as AI-generated or human-written.

The authors test their approach on ChatGPT / GPT4 / LLAMA-2 generated responses to prompts from Reddit. Using 5-10 exemplars, the authors find their approach to be quite effective at: 1) detecting whether a text was generated by a particular LM; 2) whether the text was generated by any of the held-out LLMs vs human-written.

**Strengths:**

1. Given the growing risk of plagiarism in college essays and the spread of misinformation using large language models, the topic of the paper is timely and very important.

2. The proposed algorithm in the paper is interesting and based on an intuitive idea: LLM generated text has a different style from human-written text. The authors use an interesting contrastive learning objective to learn a strong style classifier, taking care to ensure the representations are topic-agnostic.

3. The authors use a nice variant of AUC to stress test their algorithm: restricting the AUC range to 0-1%. This is a good metric to evaluate AI-generated text detection algorithms, since the desired false-positive rate is likely to be very low in real-world scenarios.

4. The authors present several various different experimental setups of their few-shot classifier (each of which have practical significance). For instance, the authors study both a) detecting text written by a particular LLM; and b) detecting text written by any LLM. The authors also conduct ablation studies on variants of the style classifier.

**Weaknesses:**

I have few main concerns with the set of experiments conducted in the paper, which are preventing me from recommending acceptance.

1. *No external baselines*: The experimental results provide no comparisons with existing AI-generated text detection methods like DetectGPT, watermarks, GPTZero, etc [1, 2, 3]. Without these comparisons, it is hard to ground the proposed method in existing literature, and understand how it competes with alternatives. I recommend the authors to measure the performance of at least watermarks [1] and DetectGPT [2] on the Reddit datasets considered in this work.

2. *No experiments measuring robustness to paraphrasing attacks*: It is now well-established that AI-generated text detectors are very brittle against paraphrasing attacks [4, 5], including DetectGPT, watermarks and GPTZero. How robust is the proposed few-shot style classifier against paraphrasing attacks like DIPPER [4, 5]? Given the vulnerability to paraphrasing, it is critical to evaluate AI-generated detectors on this dimension. Moreover, prior work [6] has shown that paraphrasing strips out stylistic elements from text. This might make the proposed approach even more vulnerable to paraphrasing attacks.

3. Minor, but the writing of Section 3 and 4 was confusing (please see the questions below).

[1] - https://arxiv.org/abs/2301.10226
[2] - https://arxiv.org/abs/2301.11305
[3] - https://gptzero.me
[4] - https://arxiv.org/abs/2303.11156
[5] - https://arxiv.org/abs/2303.13408
[6] - https://arxiv.org/abs/2010.05700

**Questions:**

1. After reading the paper, I was a bit confused by the first sentence in Section 3.1, "The experiments in this paper deal with both the supervised and few-shot learning regimes". Where exactly were the experiments in the fully supervised regime? Please correct me if I'm wrong, but to the best of my understanding, the model was trained on data from "Amply Available and Cheap" LLMs (like GPT2), and then tested on harder LLMs (like ChatGPT) using few-shot style matching on this trained model.

2. I was a bit confused in Section 3.2 on the procedure to build the style representation model. Was the UAR model used as the pretrained base model? Was the recipe inspired by Wegmann et al. (2022) used on top of the UAR model? or were these two different models? In which model was the AAC dataset used?

3. Please mention that ChatGPT / GPT4 / LLAMA-2 were used to create the LWD data in the main body of the paper. This is a crucial experimental detail, and should not be in the Appendix.

---

> ### Author Response · Authors · 2023-11-23
> **Author response**
>
> Thank you for the thoughtful comments! We first respond to the main concerns.
>
> > No external baselines: The experimental results provide no comparisons with existing AI-generated text detection methods like DetectGPT, watermarks, GPTZero, etc
>
> We do not emphasize zero-shot baselines in the original draft for two reasons. First, we evaluate a few-shot setting, and in general we expect methods that avail of the few-shot sample to perform better. In other words, it seems unfair to pit zero-shot methods against few-shot methods. Nonetheless, we agree that further comparisons would only strengthen our contribution, and so we include additional zero-shot baselines in the updated paper. Specifically, we compare our approach to additional standard approaches, including watermarking and three common detectors:
>
> * Rank - https://arxiv.org/pdf/1906.04043.pdf
> * LogRank - https://arxiv.org/ftp/arxiv/papers/1908/1908.09203.pdf
> * Entropy - https://aclanthology.org/2020.acl-main.164.pdf
>
> We additionally compare against OpenAI’s “off-the-shelf” pre-trained detector, in addition to our re-trained version of it. We find that all these methods perform worse than the proposed stylistic approaches. These experiments are shown in Table 1 with comparisons to watermarking in Appendix G and described below.
>
> We did not evaluate DetectGPT or GPTZero for the following reasons:
>
> * DetectGPT performs 100 perturbations per sample using a 3-billion T5 model. In general, we view this as a serious limitation of the approach, and in practice we lack the computational resources to evaluate this method in a reasonable amount of time.
> * We found the API cost for GPTZero to be too high, and for the size of our datasets and number of evaluations we would’ve hit the maximum token limit. Furthermore, since this is a commercial system, we lack information to reproduce any results without going through the online API, which further limits our ability to control the training data of the approach.
>
> For watermarking, we compare our detection method to the approach proposed by Kirchenbauer et al. (2023). However, before reporting results, we reiterate our original justification for not including this comparison in our original submission: adversaries are not interested in watermarking their LLM outputs for easy detection. Thus we include this comparison only as an interesting additional result; watermarking is not a practical solution to defeating sophisticated adversaries in many domains of concern.
>
> As pointed out by the reviewer, watermarking based approaches are vulnerable to simple paraphrasing attacks, we replicate this phenomenon and additionally demonstrate our approach’s robustness to the attack. To evaluate watermarking capabilities, we create an additional watermarked dataset of 1000 Amazon review generations sampled from Llama2-7B. To simulate a simple attack on this watermarked data, we paraphrase each document using DIPPER, setting the lexical diversity hyperparameter to 20%.
>
> The table below shows watermark detection results (under the AUC’ metric), we vary the number of tokens in the sample to observe performance changes as evidence increases. Similar to previous work, we find that watermarking degrades significantly even with small edits to text. We find that our proposed approach is robust to paraphrasing.
>
> | Number of Tokens | Watermarking (Original) | Watermarking (Paraphrased) | UAR - 5M (Original) | UAR - 5M (Paraphrased) |
> | ---------------- | ----------------------- | -------------------------- | ------------------- | ---------------------- |
> | 16               | 0.689                   | 0.507                      | 0.523               | 0.512                  |
> | 32               | 0.807                   | 0.504                      | 0.667               | 0.629                  |
> | 48               | 0.895                   | 0.503                      | 0.848               | 0.793                  |
> | 64               | 0.93                    | 0.505                      | 0.958               | 0.929                  |
> | 80               | 0.949                   | 0.506                      | 0.983               | 0.965                  |
> | 96               | 0.969                   | 0.5                        | 0.997               | 0.991                  |
> | 112              | 0.975                   | 0.5                        | 0.999               | 0.998                  |
> | 128              | 0.982                   | 0.502                      | 0.999               | 0.998                  |
> | 144              | 0.989                   | 0.5                        | 0.999               | 0.995                  |
> | 160              | 0.988                   | 0.504                      | 1                   | 0.999                  |

---

> ### Author Response · Authors · 2023-11-23
> **Author response (continued)**
>
> > No experiments measuring robustness to paraphrasing attacks: It is now well-established that AI-generated text detectors are very brittle against paraphrasing attacks [4, 5], including DetectGPT, watermarks and GPTZero. How robust is the proposed few-shot style classifier against paraphrasing attacks like DIPPER [4, 5]? Given the vulnerability to paraphrasing, it is critical to evaluate AI-generated detectors on this dimension. Moreover, prior work [6] has shown that paraphrasing strips out stylistic elements from text. This might make the proposed approach even more vulnerable to paraphrasing attacks.
>
> Thank you for the interesting question. The proposed framework can account for paraphrasing attacks in two ways:
>
> 1. The few-shot example may itself consist of a handful of examples of paraphrased text, in which case we can identify further instances of paraphrased LLM text as usual.
> 2. If the test setting features both LLM text and paraphrased LLM text, our multi-LLM detection approach may be applied in which, for example, a few-shot sample of GPT-4 and paraphrased GPT-4 are used to build the LLM detector.
>
> We expect the second approach to provide additional robustness since an attacker may vary the degree of paraphrasing. To validate this hypothesis, we conduct an additional experiment in which DIPPER (Krishna et al., 2023), https://arxiv.org/pdf/2303.13408.pdf is used to paraphrase LLM text.
>
> We test two versions of UAR, one where the support examples come strictly from the original LLM, and another that mimics the Multi-LLM scenario in that we provide UAR with one support example of the original LLM and another of the paraphrased LLM. For both experiments, we vary the proportion of queries that are paraphrased from 0% to 100% and set N=5.
>
> |                 |      0.0 |     0.25 |      0.5 |     0.75 |      1.0 |
> |:----------------|---------:|---------:|---------:|---------:|---------:|
> | UAR (Reddit 5M) | **0.90446**  | **0.910341** | 0.86506  | 0.797051 | 0.706332 |
> | UAR Multi-LLM   | **0.90446**  | 0.907958 | **0.890054** | **0.881983** | **0.886131** |
> | PROTONET        | 0.871244 | 0.865504 | 0.81867  | 0.752616 | 0.666963 |
>
> We observe that both UAR (Reddit 5M) and PROTONET suffer as the proportion of paraphrased queries increases. However, including the paraphrased LLM as a support example (UAR Multi-LLM) ameliorates the drop in performance.
>
> Regarding the additional questions:
>
> > After reading the paper, I was a bit confused by the first sentence in Section 3.1, "The experiments in this paper deal with both the supervised and few-shot learning regimes". Where exactly were the experiments in the fully supervised regime? Please correct me if I'm wrong, but to the best of my understanding, the model was trained on data from "Amply Available and Cheap" LLMs (like GPT2), and then tested on harder LLMs (like ChatGPT) using few-shot style matching on this trained model.
>
> We agree that this should be clarified in the main text and have made appropriate revisions. To confirm, the “zero-shot” baseline, namely “RoBERTa (zero-shot)”, employs a supervised training regime. We argue in the paper that such supervised zero-shot approaches are fundamentally limited due to distribution shifts in real-world data (e.g., introduction of new LLM or new domain relative to training data), which is why we emphasize few-shot approaches and baselines. We show the brittleness of the supervised training regime in Appendix A, where we evaluate what happens to supervised detectors as the testing distribution shifts.
>
> > Was the UAR model used as the pretrained base model? Was the recipe inspired by Wegmann et al. (2022) used on top of the UAR model? or were these two different models? In which model was the AAC dataset used?
>
> They are two different models. Both the CISR method of Wegmann et al. (2022) and the UAR approach of Rivera-Soto et al. (2021) are evaluated in our experiments. We additionally propose extensions of UAR for our machine-text detection setting, such as training on additional machine “authors,” which outperforms in certain settings such as the multi-LLM setting of S4.5. We focus on UAR since the training recipe is more generally applicable, as it does not require mining positive and negative pairs based on topic labels like CISR (which aren’t always available), and we find consistent benefits in machine-text detection performance by training on larger corpora of human-authored text. Note that the additional human-text only domains we train on (Twitter and StackExchange) do not contain topic labels for the CISR training approach.
>
> > Please mention that ChatGPT / GPT4 / LLAMA-2 were used to create the LWD data in the main body of the paper. This is a crucial experimental detail, and should not be in the Appendix.
>
> Thanks for the suggestion! While we highlight these LLMs in Figure 1(b), we agree their inclusion in LWD should be emphasized in the main text and have made the appropriate revisions in the draft.

---

> > ### Comment · Reviewer_pncH · 2023-12-01
> > **Thank you, raising the score to 6**
> >
> > Thank you for your detailed comments! I've raised my score to the accept range of 6.

---

### Official Review · Reviewer_7sQm · 2023-10-27

**Soundness:** 2 fair
**Presentation:** 3 good
**Contribution:** 2 fair
**Rating:** 5
**Confidence:** 4

**Summary:**

Few-Shot Detection of Machine-Generated Text using Style Representations
The paper introduces a novel method to differentiate human- and LLM-generated text. The method learns style representations from data that is labeled with an author. Afterward, these style representations are extracted from human- and LLM-generated text to obtain a fixed-size representation that is then used to detect LLM-written texts. In particular, the few-shot setting is analyzed where a few "support samples" are available for the LLM(s) in question where the extracted style representations are then compared to some "query samples" to classify whether they were written by the LLM(s) in question. The similarity between support and query samples is used to determine whether an LLM wrote the text.

**Strengths:**

In general, the paper is well-written. The paper tackles a relevant problem: Detecting LLM-written text. Previous work, such as the classifier released by OpenAI, had much worse performance. Also, the approach is new: Instead of supervised training, the method effectively uses pre-trained style representations for the few-shot setting. The results of the experiments look convincing (maybe overly convincing, considering that, for instance, in Table 1, AUC' is almost perfect even for a false positive rate of <= 1%?).

**Weaknesses:**

I tend to reject the paper. The reasons are the following: 1. In the end, the paper boils down to applying a few methods that learn stylistic and semantic representations to classify a document as human- or LLM-written. There is no new method, and the contribution lies in applying existing methods and creating a training and evaluation corpus for AAC and LWD LLMs (what the authors do not mention as a contribution, and these datasets will not be made available according to the paper). 2. The experiments miss some details, and it is unclear why the proposed UAR is trained on different datasets than the other methods.

**Questions:**

How is the similarity score between support and query samples calculated?
Section 4.1:
Where is the Amazon dataset coming from? How is it used? Also, in the multi-domain variation?
Section 4.3:
The metric being reported is AUC'. This is rather uncommon. Why not report the normal AUC? Maybe all methods look too similar, then? Also, instead of introducing a new AUC' the authors could have resorted to something more in line with other works, E.g., something similar to FPR95 but reflecting the importance of a low false positive rate. Using the reasoning from section 4.3, a TPR95 (also not in the literature like AUC', but would be more in line with FPR95.). Missing details in the experiment section: Size of training and evaluation corpora, which LLMs are analyzed. It would help if the authors also mentioned that AUC' is normalized.
Section 4.4:
The paper mentions that Reddit is the main training corpus and, therefore, not used for evaluation. This contradicts section 4.1, where Reddit is also in the evaluation corpus. So, how is Reddit used?
Table 2 shows different methods trained on different training corpora. Since the paper emphasizes that UAR is THE method to perform few-shot LLM text detection, why are the baselines not trained on the same training corpora?
What is the difference between Reddit (5M) and Reddit (used for SBERT) in Table 2
Why is Twitter from Table 2 not mentioned in Table 1?
How do the authors explain that the proposed UAR (based on SBERT) is still much better than SBERT? The authors emphasize that SBERT is unsuitable as it is trained to capture semantics. So why not use a "normal" BERT in UAR?
RoBERTa (zero shot) is not introduced. What is it exactly?
Section 4.5:
The text mentions "other UAR variants." Are the Multi-LLM and Multi-domain models UAR variants? This should be clarified. Also, again, the same problem as in the other sections: why not train the other methods with this setup?


Additional remarks:
-Section 4.5, first paragraph, last row: "the number of posts" should probably be "the number of tokens" or are the descriptions and axis titles wrong in Figure 2?

---

> ### Author Response · Authors · 2023-11-23
> **Author response**
>
> Thanks for the feedback! We first address the main concerns.
>
> > the contribution lies in applying existing methods and creating a training and evaluation corpus for AAC and LWD LLMs (what the authors do not mention as a contribution, and these datasets will not be made available according to the paper).
>
> We do intend to release all the datasets mentioned in the paper (the reproducibility statement has been updated to clarify this point). Regarding the novelty of the approach, to the best of our knowledge, few-shot detection of machine-generated text using style representations is a new method:
>
> * We are the first to systematically explore few-shot machine detection. We argue that this is necessary in light of real-world distribution shifts to obtain a robust machine-text detector. This is a non-trivial departure from prior work in machine-text detection and offers a practical way to tackle distribution shifts pervasive in most settings where one may be interested in detecting machine-generated text.
> * We are the first to find that representations of writing style estimated solely from human-written text (i.e., negative examples) can outperform state of the art few-shot machine-text detection methods, even though those baseline methods are trained using additional positive examples of machine-generated texts (Table 2). We argue that this is a significant finding in light of the proliferation of LLMs, since it is impractical to repeatedly train new supervised classifiers on the basis of the latest LLMs. Our approach opens the door to building “training-free” detectors for the latest LLMs that are tailored to a variety of downstream settings (e.g., education, science, social media).
>
> We also explore several simple but important adaptations of existing style representations for our machine-text detection task, rather than simply applying them “off-the-shelf”, including training on significantly more data, training on combinations of machine-generated and human-generated data, and applying them to several detection settings (e.g., multi-LLM).
>
> > The experiments miss some details, and it is unclear why the proposed UAR is trained on different datasets than the other methods.
>
> The proposed approach is trained on different data than the non-stylistic baselines because it does not require examples of machine generated text. This is a key finding of our work: representations trained only on human-authored text (e.g., Reddit, Twitter, StackExchange) yield stylistic representations that can effectively discriminate between human and machine authors.

---

> ### Author Response · Authors · 2023-11-23
> **Author response (continued)**
>
> Thank you for the detailed questions. We have taken the opportunity to clarify these details in the main text and in the appendix, but also provided an exhaustive response below.
>
> > How is the similarity score between support and query samples calculated?
>
> We compute the cosine similarity between the support embedding and the query embedding as outlined in the first paragraph of Section 3.2.
>
> > Section 4.1: Where is the Amazon dataset coming from? How is it used? Also, in the multi-domain variation?
>
> As stated in the reproducibility statement, the Amazon dataset comes from: https://nijianmo.github.io/amazon/index.html#files. It is strictly used to generate LWD data that is used for evaluation. We do not use it as training data.
>
> > Section 4.3: The metric being reported is AUC'. This is rather uncommon. Why not report the normal AUC? Maybe all methods look too similar, then? Also, instead of introducing a new AUC' the authors could have resorted to something more in line with other works, E.g., something similar to FPR95 but reflecting the importance of a low false positive rate. Using the reasoning from section 4.3, a TPR95 (also not in the literature like AUC', but would be more in line with FPR95.).
>
> We choose to emphasize AUC’ since we focus on an operating point with a low false alarm rate, which we argue is necessary for real-world applications of machine-text detectors. Our findings are consistent with other metrics at other operating points. Below (and also in the updated draft in Appendix H, Table 15), we show results when we truncated every evaluation document to the nearest sentence boundary before the thirty-second token:
>
>  | **Method** | **Training Dataset** | **AUC** | **AUC'** | **FPR@95TPR** |
>  |     :----:    |     :----:    |     :----:    |     :----:    |     :----:    |
>  | **Few-Shot Methods** |  |  |  |
>  | LUAR  |  Reddit(5M)  |  **0.9884** | 0.9094 |  **0.0533** |
>  | LUAR  |  Reddit (5M), Twitter, StackExchange  |  **0.9884** |  **0.9118**  | 0.0543 |
>  | LUAR  |  AAC, Reddit (politics)  | 0.8913 | 0.7308 | 0.3394 |
>  | CISR  |  Reddit (hard neg/hard pos)  | 0.9608 | 0.7973 | 0.144 |
>  | RoBERTa (ProtoNet)  |  AAC, Reddit (politics)  | 0.9791 | 0.9062 | 0.0934 |
>  | RoBERTa (MAML)  |  AAC, Reddit (politics)  | 0.6686 | 0.5141 | 0.6555 |
>  | SBERT  |  SNLI, MultiNLI  | 0.9673 | 0.8087 | 0.1448 |
>  | **Zero-Shot Methods** |  |  |  |  |
>  | RoBERTa (Zero shot)  |  AAC, Reddit (politics)  | 0.7238 | 0.5295 | 0.5369 |
>  | OpenAI Detector  |  WebText, GPT2-XL  | 0.6933 | 0.5559 | 0.7698 |
>  | Rank (GPT2-XL)  |  BookCorpus, WebText  | 0.7301 | 0.5423 | 0.6341 |
>  | LogRank (GPT2-XL)  |  BookCorpus, WebText  | 0.9107 | 0.6395 | 0.2209 |
>  | Entropy (GPT2-XL)  |  BookCorpus, WebText  | 0.2083 | 0.4977 | 0.9667 |
>  | Random |  | 0.50 | 0.05 | -- |
>
> When evaluating using smaller writing samples and looking at standard metrics, the proposed style-based approaches continue to outperform both few-shot baselines and standard zero-shot baselines.
>
> > Missing details in the experiment section: Size of training and evaluation corpora, which LLMs are analyzed.
>
> The details for each of the datasets are mentioned in the Appendix due to the page limit; we agree that having this in the main text would be helpful. Appendix C shows the statistics of the datasets used to train UAR, while Appendix E covers the details of creating the AAC and LWD datasets as well as their respective statistics and the details of the M4 dataset. We evaluate all our approaches on unseen domains and unseen LLM’s including Llama-2, ChatGPT, GPT-4, Dollyv1, Dollyv2, Davinci, FlanT5, Cohere, and BloomZ.
>
> > It would help if the authors also mentioned that AUC' is normalized.
>
> We have clarified this point in paper.

---

> ### Author Response · Authors · 2023-11-23
> **Author response (continued)**
>
> > Section 4.4: The paper mentions that Reddit is the main training corpus and, therefore, not used for evaluation. This contradicts section 4.1, where Reddit is also in the evaluation corpus. So, how is Reddit used? Table 2 shows different methods trained on different training corpora. Since the paper emphasizes that UAR is THE method to perform few-shot LLM text detection, why are the baselines not trained on the same training corpora?
>
> We made an error in Section 4.1 when we mentioned that Reddit was used as an evaluation corpora. The evaluation datasets are Amazon, M4 PeerRead, M4 ArXiv, M4 WikiHow, and M4 Wikipedia. The Reddit corpora is used strictly for training, which enables us to evaluate robustness to distribution shift.
>
> For baselines which require AAC data (MAML, Protonet, and RoBERTa), we train on a politics-only version of the Reddit dataset. We restrict the topic to politics to ensure that these baselines learn features that are agnostic to the domain. In Appendix B, we perform ablations with ProtoNet where we show that controlling the domain is essential to achieving good results.
>
> As mentioned earlier, the proposed style-based representation can be trained using only negative (human-written) examples, and as a result we can train on more datasets such as StackExchange.
>
> > What is the difference between Reddit (5M) and Reddit (used for SBERT) in Table 2
>
> Thanks for catching this; the original text is inaccurate in this case. We used an off-the-shelf SBERT, specifically paraphrase-distilroberta-base-v1 which is actually trained on a variety of domains, including some of our domains intended to be held-out (such as Wikipedia). This is strictly to the advantage of the SBERT baseline since it suffers less from distribution shifts, while we control the training data of our stylistic representations to ensure the held-out evaluation domains do introduce distribution shifts. In other words, our baseline is given an unfair advantage relative to our proposed methods, yet we still outperform it, which we attribute to the stylistic training objective.
>
> >  Why is Twitter from Table 2 not mentioned in Table 1?
>
> Twitter is only used as a training corpus for the Multi-LLM UAR. This is the reason it was not mentioned in Table 1 in the original draft.
>
> > How do the authors explain that the proposed UAR (based on SBERT) is still much better than SBERT? The authors emphasize that SBERT is unsuitable as it is trained to capture semantics. So why not use a "normal" BERT in UAR?
>
> The proposed UAR was trained to capture the stylistic differences that are agnostic to the topic being written about. SBERT was trained to capture the semantic similarity, hence its embedding intrinsically captures the topic of a piece of text. The stylistic features of LLMs remain quite stable across the various domains considered, hence the effectiveness of the UAR in capturing the differences in writing styles between different LLMs. The reason to initialize UAR with SBERT over BERT is simply that SBERT was also trained with a contrastive objective, although in practice we suspect that initializing from BERT would perform equally well since the UAR training data is very large (hundreds of millions of documents) and hence the initialization is less critical.
>
> > RoBERTa (zero shot) is not introduced. What is it exactly?
>
> RoBERTa (Zero-Shot) shares the same architecture as the OpenAI Detector except that it is trained on the Reddit+AAC corpora. As a reminder, we do this to introduce a distribution shift at evaluation time, and the proposed stylistic representations are also limited to training on disjoint domains rather than the evaluation domains. Elsewhere in this response and in Table 1 in the updated draft, we additionally report results using the pre-trained OpenAI model, which was trained on a variety of domains, likely including our evaluation domains. Nonetheless, we find that it performs worse than our “homegrown” version.
>
> > Section 4.5: The text mentions "other UAR variants." Are the Multi-LLM and Multi-domain models UAR variants? This should be clarified. Also, again, the same problem as in the other sections: why not train the other methods with this setup?
>
> Yes, they are UAR variants, which was mentioned in Section 3.2 of the original paper under the “Multi-domain” and “Multi-LLM” headers.

---

> > ### Comment · Reviewer_7sQm · 2023-12-01
> >
> > Thank you for the detailed feedback. The authors have addressed one of our major concerns, the AUC' metric. However, the way it was addressed means that the extended AUC vs AUC' results are not comparable to the AUC vs AUC' results in the originally proposed setup, which would have been more interesting. The setting has now been changed to smaller samples that are harder to detect, making the extremely high AUC' scores from the simpler setup not comparable to this more challenging setup. Additionally, the normalization of AUC is still not mentioned.
> > The authors also argue that cosine similarity is mentioned in the first paragraph of Sec 3.2, which is not the case (just angular similarity). Furthermore, the authors mention the number of parameters for some models but not for SBERT. As it is a "distil" variant, it is likely much smaller than UAR. Inconsistencies in the number of digits have been introduced in Table 1. Overall, the submitted score, therefore, reflects our previous rating.

---

### Official Review · Reviewer_f5Yg · 2023-10-27

**Soundness:** 2 fair
**Presentation:** 3 good
**Contribution:** 2 fair
**Rating:** 3
**Confidence:** 5

**Summary:**

The paper proposes a new approach to detecting machine-generated text using style representations. The authors leverage representations of writing style estimated from human-authored text to effectively distinguish between human and machine-generated text. They demonstrate the effectiveness of their approach through experiments on several datasets and show that their method outperforms previous approaches to detecting machine-generated text.

Overall, the paper presents a novel and potentially promising approach to detecting machine-generated text that could have important implications for identifying and mitigating the risks associated with the use of language models that convincingly mimic human writing. However, the work has strong limitations in training the style representation learning network. Moreover, the comparison to existing approaches is not sufficient.

**Strengths:**

Pros:


- The proposed approach leverages representations of writing style estimated from human-authored text, which can effectively distinguish between human and machine-generated text.

- The approach does not rely on samples from language models of concern at training time, which makes it more robust to data shifts and more practical to implement.

- The experiments conducted by the authors demonstrate that their approach outperforms previous approaches to detecting machine-generated text.

**Weaknesses:**

Cons:

- The approach assumes that documents generated from amply available and cheap (AAC) models are available at training time, which may not always be the case in practice.

- The work has strong limitations in training the style representation learning network. Basically, the writing style can also be changing over time. The existing writing style training dataset may not be powerful enough to cover the whole. The work adopts(and highly depends on) a supervised approach for this, which has many limitations in real applications.

- Moreover, the comparison to existing approaches is not sufficient. More baselines are supposed to be included. Here are the most recent surveys on it. Other representative works are suggested to be included, such as OpenAI's detector and other training-free approaches should be compared because they do not have domain adaption issues.
 https://arxiv.org/pdf/2310.15264.pdf

https://arxiv.org/pdf/2310.15654.pdf

https://arxiv.org/pdf/2310.14724.pdf

- The authors should consider additional qualitative and quantitative metrics to evaluate the specification tasks. The paper adopts only AUROC as the metric. However, others such as F1 and TPR are more reasonable to use. Its is important to consider additional metrics to ensure consistency of performance especially in leveraging these algorithms for a product use case.  The models could be compared based on a perplexity score-based approach to help find the disambiguation between human and ChatGPT-generated text. Here is a reference paper https://arxiv.org/pdf/2301.13852.pdf

- The approach may not be effective against more advanced language models that can better mimic human authorship.

**Questions:**

Questions:

- The authors should consider additional qualitative and quantitative metrics to evaluate the specification tasks. The paper adopts only AUROC as the metric. However, others such as F1 and TPR are more reasonable to use. Its is important to consider additional metrics to ensure consistency of performance especially in leveraging these algorithms for a product use case.  The models could be compared based on a perplexity score-based approach to help find the disambiguation between human and ChatGPT-generated text. Here is a reference paper https://arxiv.org/pdf/2301.13852.pdf

---

> ### Author Response · Authors · 2023-11-23
> **Author response**
>
> Thanks for reviewing our paper! We have prepared a detailed response which we hope address your questions and concerns. Please also refer to our general response for further experiments requested by other reviewers.
>
> First of all, as mentioned in our response to all reviewers, we do intend to release the datasets we created and have clarified this point in our reproducibility statement. We note that such datasets are already amply available publically, and in fact, as mentioned in S4.1, the publicly available M4 dataset comprises a portion of our evaluation. For this reason we disagree with the reviewer’s assertion that AAC data may be unavailable in practice, unless the reviewer was referring to AAC data in low-resource languages, in which case we agree but would argue that there are fewer LLM of concern in such languages. In any case, for the sake of reproducing our results exactly, we agree that the datasets we generated must be released.
>
> > The work has strong limitations in training the style representation learning network. Basically, the writing style can also be changing over time. The existing writing style training dataset may not be powerful enough to cover the whole.
>
> We suspect that there may be a misunderstanding concerning the nature of the writing style representations used in this work. Such representations are specifically trained to capture temporally-invariant aspects of writing style and their ability to generalize to future data has been validated in prior work (see e.g. https://arxiv.org/abs/1910.04979). For example, as shown in our experiments, such representations are robust to various distribution shifts arising from the introduction of new domains relative to the training data used to train the style representations. Furthermore, as illustrated in Figure 1, LLM-generated text, even when produced using prompts designed to elicit varied writing styles, exhibits a consistent stylistic marker. This suggests that LLM-generated text indeed contains temporally-invariant stylistic traits that enable our approach to successfully detect them, perhaps a consequence of the fact that the LLM parameters are frozen.
>
> > [...] the comparison to existing approaches is not sufficient. More baselines are supposed to be included. Here are the most recent surveys on it. Other representative works are suggested to be included, such as OpenAI's detector and other training-free approaches should be compared because they do not have domain adaptation issues.
>
> Given our new few-shot setting for machine-text detection, it seemed appropriate for us to focus our comparison on few-shot baselines, since these use the same data as the proposed approach. Therefore, the original paper includes comparisons to ProtoNet and MAML, two standard few-shot learning methods, and several additional few-shot detectors based on pre-trained representations including SBERT and CISR. Nevertheless, as mentioned in our general response to all reviewers, we have now included an additional four “zero-shot” baseline models, including OpenAI’s detector and training-free approaches, in the experiment reported in Table 1 of the revised paper, as well as comparisons to watermarking in Appendix G (Figure 4). We find that the proposed approach significantly outperforms all other training-free approaches.
>
> Regarding reporting additional standard metrics, we report additional experiments in Appendix I on smaller writing samples (making the detection problem harder), where we include standard metrics. These results are reproduced below for convenience.
>  | **Method** | **Training Dataset** | **AUC** | **AUC'** | **FPR@95TPR** |
>  |     :----:    |     :----:    |     :----:    |     :----:    |     :----:    |
>  | **Few-Shot Methods** |  |  |  |
>  | LUAR  |  Reddit(5M)  |  **0.9884** | 0.9094 |  **0.0533** |
>  | LUAR  |  Reddit (5M), Twitter, StackExchange  |  **0.9884** |  **0.9118**  | 0.0543 |
>  | LUAR  |  AAC, Reddit (politics)  | 0.8913 | 0.7308 | 0.3394 |
>  | CISR  |  Reddit (hard neg/hard pos)  | 0.9608 | 0.7973 | 0.144 |
>  | RoBERTa (ProtoNet)  |  AAC, Reddit (politics)  | 0.9791 | 0.9062 | 0.0934 |
>  | RoBERTa (MAML)  |  AAC, Reddit (politics)  | 0.6686 | 0.5141 | 0.6555 |
>  | SBERT  |  SNLI, MultiNLI  | 0.9673 | 0.8087 | 0.1448 |
>  | **Zero-Shot Methods** |  |  |  |  |
>  | RoBERTa (Zero shot)  |  AAC, Reddit (politics)  | 0.7238 | 0.5295 | 0.5369 |
>  | OpenAI Detector  |  WebText, GPT2-XL  | 0.6933 | 0.5559 | 0.7698 |
>  | Rank (GPT2-XL)  |  BookCorpus, WebText  | 0.7301 | 0.5423 | 0.6341 |
>  | LogRank (GPT2-XL)  |  BookCorpus, WebText  | 0.9107 | 0.6395 | 0.2209 |
>  | Entropy (GPT2-XL)  |  BookCorpus, WebText  | 0.2083 | 0.4977 | 0.9667 |
>  | Random |  | 0.50 | 0.05 | -- |
>
> Note that Rank, LogRank, and Entropy are standard methods from the literature:
> * Rank - https://arxiv.org/pdf/1906.04043.pdf
> * LogRank - https://arxiv.org/ftp/arxiv/papers/1908/1908.09203.pdf
> * Entropy - https://aclanthology.org/2020.acl-main.164.pdf

---

> ### Author Response · Authors · 2023-11-23
> **Author response**
>
> > The approach may not be effective against more advanced language models that can better mimic human authorship.
>
> The reviewer’s concern about the robustness of our approach to future, more fluent language models was precisely the motivation for our research methodology, which simulates precisely this situation. Indeed, we chose to evaluate documents by LWD models, which produce more fluent text than the AAC models used at training time for some proposed variations of style representations, and were developed in the future relative to those models. We also note that the proposed variation of style representations not availing of any machine generated data still detects LWD models better than baseline models in many cases (see Table 1 in the revised paper, or the reproduced table above), some of which are trained with AAC.

---

> > ### Comment · Reviewer_f5Yg · 2023-11-23
> > **Thanks for the response**
> >
> > Thanks a lot for the response. I would like to keep my original score considering the strong limitations in training the style representation learning network and the limited comparison to zero-shot approaches such as the DetectGPT and DNA-GPT.

---

### Official Review · Reviewer_4qF2 · 2023-10-31

**Soundness:** 3 good
**Presentation:** 2 fair
**Contribution:** 3 good
**Rating:** 6
**Confidence:** 4

**Summary:**

This paper tackles the problem of detecting machine-generated text produced by LLMs. For this the authors leverage a method to extract stylistic features for authorship detection, and use those features in a nearest-neighbour algorithm to attribute a given document to either human or machine. Variants of the methods are presented, depending if the task is to detect human/machine or human/specific-LLMs.
Somehow surprisingly and contrary to previous work, this specific combination of approach and setting makes the problem tractable and produces very good results (>0.9 AUC).
The embeddings are computed with a training data of text generated with "small" and less powerful LLMs (GPT2, smaller versions of GPT2) but the evaluation is done on current SOTA models.

**Strengths:**

It presents a valuable contribution to an important topic. This is both with respect to the dataset (which hopefully will be released), as well as to the methodology. The setting of few-shot is also more realistic than previous proposals which assumed access to a very large corpus of machine-generated data.

**Weaknesses:**

The exact setting is not clear, and the presentation of the results under a single value (AUC) makes it hard to assess the impact of this work. In particular, it would be good to see some more examples to get a better intuition. It would seem like the evaluation examples are easily detectable because the LLM mimics the provided persona so well as to insist on it in the generated text (`...as a <persona>, I...` . This might be the reason for the comparably high scores (as opposed to previous work, which generated non-persona guided text instead).

The field of detecting machine-generated text is still nascent, but this paper seems to miss some relevant references:

* Daphne Ippolito, Daniel Duckworth, Chris CallisonBurch, and Douglas Eck. 2020. Automatic detection of generated text is easiest when humans are fooled.

* Antonis Maronikolakis, Mark Stevenson, and Hinrich Schutze. 2020. Transformers are better than humans at identifying generated text.

* Matthias Gallé, Jos Rozen, Germán Kruszewski, and Hady Elsahar. 2021. Unsupervised and Distributional Detection of Machine-Generated Text

**Questions:**

Will the datasets be released as well to facilitated reproducibility?

---

> ### Author Response · Authors · 2023-11-23
> **Author response**
>
> Thanks for the feedback!
>
> Our previous use of the AUC’ metric may have insinuated that the approach is only relevant for settings requiring low false positive rates. However, this is not the case, as demonstrated in the additional results reported in Appendix H and reproduced below. Here, we vary the writing sample size to yield a harder detection problem, and additionally report the standard AUC and FPR@95TPR metrics. Note that the proposed style representation methods outperform all competing methods for these standard metrics, including the additional zero-shot baselines.
>  | **Method** | **Training Dataset** | **AUC** | **AUC'** | **FPR@95TPR** |
>  |     :----:    |     :----:    |     :----:    |     :----:    |     :----:    |
>  | **Few-Shot Methods** |  |  |  |
>  | LUAR  |  Reddit(5M)  |  **0.9884** | 0.9094 |  **0.0533** |
>  | LUAR  |  Reddit (5M), Twitter, StackExchange  |  **0.9884** |  **0.9118**  | 0.0543 |
>  | LUAR  |  AAC, Reddit (politics)  | 0.8913 | 0.7308 | 0.3394 |
>  | CISR  |  Reddit (hard neg/hard pos)  | 0.9608 | 0.7973 | 0.144 |
>  | RoBERTa (ProtoNet)  |  AAC, Reddit (politics)  | 0.9791 | 0.9062 | 0.0934 |
>  | RoBERTa (MAML)  |  AAC, Reddit (politics)  | 0.6686 | 0.5141 | 0.6555 |
>  | SBERT  |  SNLI, MultiNLI  | 0.9673 | 0.8087 | 0.1448 |
>  | **Zero-Shot Methods** |  |  |  |  |
>  | RoBERTa (Zero shot)  |  AAC, Reddit (politics)  | 0.7238 | 0.5295 | 0.5369 |
>  | OpenAI Detector  |  WebText, GPT2-XL  | 0.6933 | 0.5559 | 0.7698 |
>  | Rank (GPT2-XL)  |  BookCorpus, WebText  | 0.7301 | 0.5423 | 0.6341 |
>  | LogRank (GPT2-XL)  |  BookCorpus, WebText  | 0.9107 | 0.6395 | 0.2209 |
>  | Entropy (GPT2-XL)  |  BookCorpus, WebText  | 0.2083 | 0.4977 | 0.9667 |
>  | Random |  | 0.50 | 0.05 | -- |
>
> Note that Rank, LogRank, and Entropy are standard methods from the literature:
> * Rank - https://arxiv.org/pdf/1906.04043.pdf
> * LogRank - https://arxiv.org/ftp/arxiv/papers/1908/1908.09203.pdf
> * Entropy - https://aclanthology.org/2020.acl-main.164.pdf
>
> We thank the reviewer for their keen observation about potential artifacts of our prompting scheme appearing in the LM-generated documents. This artifact occurs only with LM-generated documents in the Reddit domain, which were not used in any experiments except the one reported in Appendix A, where they were used for evaluation rather than training. Therefore the artifact does not help any model to detect machine-generated documents. We have clarified where this generated data is employed in the revised paper to avoid confusion.
> We also thank the reviewer for the references which we have included in the Related Work section of the revised paper.
> > Will the datasets be released as well to facilitate reproducibility?
>
> Yes, the data will be released. We regret not making this clear in the original reproducibility statement, which has been revised.

---

### Author Response · Authors · 2023-11-23
**General author response**

We thank all the reviewers for their helpful feedback! In response, we have uploaded a revised paper with the following additional experiments:
* Comparison with additional zero-shot baselines, which are outperformed by the proposed style representation methods (S4.4, Table 1).
* Experiments varying the writing sample size with additional metrics, including the usual AUC and FPR@95TPR, which provide further support for our initial conclusions (Appendix H, Table 15).
* Assessment of the robustness of our approach to paraphrasing attacks. We find that our approach is more robust to such attacks than baselines (Appendix I, Figure 5).
* Comparison against watermarking. We find that our detection approach is surprisingly competitive with the statistical test for detecting watermarked text for certain writing sample sizes (Appendix G, Figure 4), even though our approach is general and may also be applied in settings where adversaries decline to watermark their generated text.

As a point of clarification, we do intend to release all the datasets created as part of this work to ensure that our results are easily reproducible. The reproducibility statement has been updated to clarify this point in the revised paper.

One reviewer (pncH) felt that our proposed AUC’ metric yielded an appropriate “stress test” of our approach, while others (f5Yg, 7sQm) felt that the metric was too nonstandard. One motivation for using this metric was that when varying the size of the writing samples being evaluated, AUC tended to “saturate” sooner. Therefore, in the original submission, we used the more-stringent metric AUC’, which allowed us to vary the writing sample size without saturating the metric. Furthermore, as argued in the paper, the AUC’ metric evaluates the region of the ROC curve corresponding to a low false alarm rate, which is a prerequisite in many real-world applications of machine-text detection.

However, we recognize that different operating points may also be of interest. For this reason, we have repeated our main experiments (S4.4) using smaller writing samples, thereby making the problem harder and other metrics more informative. In Appendix H, we report both the usual AUC and our proposed AUC’ as well as FPR@95TPR, which was requested by f5Yg and 7sQm. We note that the standard AUC tracks well with AUC’ in terms of the relative performance of the various models, with the proposed stylistic representations outperforming all other methods. For example, our extension of UAR trained only on “negative” examples (i.e., human-written documents) achieves the best performance under FPR@95 in this setting, as in our original results using larger writing samples.

Several reviewers requested that we include OpenAI’s AI Detector as a baseline model. We have now included that model in the experiment reported in Table 1, along with a number of further zero-shot baselines, all of which perform worse than the proposed style representation approaches. Our rationale for originally excluding the OpenAI model was that the model listed as “RoBERTa (Zero shot)” in the original submission has the same architecture, albeit re-trained on different data. In fact, our re-trained version outperforms the off-the-shelf OpenAI model.

---

### Meta-Review · Area_Chair_NBnw · 2023-12-06

**Metareview:**

This paper focuses on few-shot detection of machine generated text. In constrast to existing work that relies on training samples from language models, the authors propose a novel approach that is based on stylistic features extracted from human-authored text. This approach is experimentally demonstrated to be effective for distinguishing machine vs human-authored text and outpeforms a range of baselines.
The reviewers have raised a number of concerns; however, these have been largely addressed in the rebuttal. Specifically, the authors have now added a number of new experiments, namely comparisons with additional stronger (zero-shot) baselines, experiments with additional metrics, an assessment of the robustness of their approach to paraphrasing attacks, and comparisons against watermarking. Overall, the paper presents a novel and promising approach to detecting machine-generated text.

**Justification For Why Not Higher Score:**

This is an interesting piece of work that tackles an important problem and that would benefit the community; however, it is not of a groundbreaking nature and, therefore, I believe it is better suited as a poster paper.

**Justification For Why Not Lower Score:**

This is an interesting piece of work that tackles an important problem with a solution that would benefit the community.

---

### Decision · Program_Chairs · 2024-01-16

Accept (poster)